# The impacts of data constraints on the predictive performance of a general process-based crop model (PeakN-crop v1.0)

Silvia Caldararu[1*], Drew W. Purves[1], and Matthew J. Smith[1]

[1]Microsoft Research, Cambridge, UK
[*]Now at Max Planck Institute for Biogeochemistry, Jena, Germany

*Correspondence to:* Matthew Smith, Matthew.Smith@microsoft.com

**Abstract.** Improving international food security under a changing climate and increasing human population will be greatly aided by improving our ability to modify, understand and predict crop growth. What we predominantly have at our disposal are either process based models of crop physiology or statistical analyses of yield datasets, both of which suffer from various sources of error. In the current paper we present a generic process based crop model which we parametrise using a Bayesian model fitting algorithm to three different sources of data - space based vegetation indices, eddy covariance productivity measurements and regional crop yields. We show that the model parametrised without data, based on prior knowledge of the parameters, can largely capture the observed behaviour but the data constrained model greatly improves both the model fit and reduces prediction uncertainty. We investigate the extent to which each dataset contributes to the model performance and show that while all data improves on the prior model fit, the satellite based data and crop yield estimates are particularly important for reducing model error and uncertainty. Despite these improvements, we conclude that there are still significant knowledge gaps, in terms of available data for model parametrisation, but our study can help indicate the necessary data collection to improve our predictions of crop yields and crop responses to environmental changes.

## 1 Introduction

Improving food security is one of the greatest challenges currently facing humanity (Schmidhuber and Tubiello, 2007; Rosegrant and Cline, 2003). The increasing and developing human population is driving up food demand and changing demand patterns. This is occurring alongside increasing anthropogenic threats to supply such as climate change. Predicting and understanding how crops respond to changes in their environment through the use of mathematical models is needed to help address such threats, enabling advanced warning of potential threats and predictions of what alterations to agricultural practices might help prevent or mitigate problems. A continual challenge when developing models is knowing the generality of their predictions, either applied to multiple crops or across different space and time scales (Rosenzweig et al., 2014). Having one model to cover all circumstances is obviously unrealistic, as is tailor-making models to every conceivable circumstance. Thus, a challenge in developing models to help address the current food security crisis is identifying those that can be said to be generally useful over particular scales of application. In the current study we present a proof of concept that such an aim can be reached through using a process-based crop model, parametrised to available data using a model fitting algorithm.

Most crop models to date can be put into one of two broad categories: process-based or statistical. Process-based crop models have some representation of the mechanisms that determine how plants grow in their formulation (e.g. Jamieson et al., 1998; Jones et al., 2003; Stackle et al., 2003). Processes included can cover crop phenology, carbon assimilation and biomass allocation responses to the internal plant state and the external environment. Such models have traditionally been specific to a particular crop, partly because of the nature of studies that employ process-based crop models, which have tended to focus on individual crops and often describe growth phases specific to a particular crop type within their formulation. However it is also partly because of the difficulty in developing generally applicable process based crop models; it can be unclear which aspects of the model formulation can be said to be general versus crop specific and obtaining data to assess model generality continues to be a challenge. Some studies have avoided making crop-specific models by using broad crop categories such as C3 and C4 crops, based on the functional plant type concept (Bondeau et al., 2007; Osborne et al., 2015). Other models group a family of crop-specific parametrisations into one single framework, which limits generality but does facilitate use across different scales and crops (Brisson et al., 2003; Stackle et al., 2003).

Statistical crop models aim to capture relationships between various predictor variables and crop properties without using any information of how such factors should be related from biology or ecology. For example, studies have predicted crop yields based on observed simple relationships between yield data and climate inputs (Lobell et al., 2011; Lobell and Field, 2007; Schlenker and Roberts, 2009); these have then been used to help understand past long-term trends in yields at large spatial scales and to make forward projections under climate change scenarios. Often statistical models are developed to be generally applicable to multiple crops and apply over multiple space and timescales, as these do not need to include any plant specific concepts.

Both the process-based and statistical approaches have their disadvantages when it comes to obtaining general insights. Process-based models have often only been shown to be applicable at the individual field scale, making it unclear if their predictions might provide information about crop responses at larger spatial scales. Process-based models can also be sensitive to chosen parameter values and formulation, which has rarely been identified as applicable over multiple crop types or locations (Challinor et al., 2009). Statistical models are limited by the extent to which the relationships they capture are useful in predicting crop properties outwith the circumstances that they have been verified for. This becomes a particularly important limitation given that one of the leading questions being addressed in food security is how different crops might grow in environments and under circumstances that we have not yet observed. For example, correlative models based on mean annual values of environmental variables are unlikely to capture the impacts of changes in extreme weather events or increases in atmospheric $CO_2$, which have been shown to be essential to understanding changes in crop yield under climate change (Porter and Semenov, 2005; Deryng et al., 2014). Furthermore, simple statistical analyses rarely incorporate information on management agricultural practices such as planting and harvest dates, irrigation and fertiliser application, which account for a large proportion of variations in yield across the globe (Calvino et al., 2003; Zwart and Bastiaanssen, 2004).

An alternative to the extremes of either purely process-based or purely statistical crop models is to apply statistical methods to process-based models to data-constrain their parameters. This technique, which is increasingly used in in earth system and vegetation modelling studies (Fox et al., 2009; Raupach et al., 2005), involves allowing some parameters to have undefined

values and inferring those values by comparing the model predictions to data (hence the technique is called parameter inference, or inverse modelling). The specific methods used vary but the aim is often commonly to deduce parameters that yield the best model predictive performance (another common aim is to deduce insight about the underlying processes from the inferred parameter values). The result is typically a model with improved model predictive ability (Knorr et al., 2010; Ziehn et al., 2012) when assessed using empirical data. Importantly, formally data-constraining model parameters is a technique that can be used to increase the general applicability of a given model formulation, and for that general applicability to be assessed.

The main problem with data-constraining process based models is data availability. Datasets of annual yield such as those used in statistical modelling studies are unlikely to be sufficient when data-constraining the parameters of physiologically explicit model because, to put it simply, they are unlikely to carry enough information to enable identification of what the different model parameters should be. However two other sources of data, widely used in the global vegetation modelling but to a lesser extent in agricultural modelling, could be of use in data constraining crop model parameters. Space based remote sensing data can provide spatially and temporally continuous information on vegetation greenness at a variety of spatial and temporal scales (Glenn et al., 2008; Tucker et al., 2005). Such data has previously been used for crop classification purposes (Wardlow et al., 2007; Howard et al., 2012) and for simple yield estimation (Doraiswamy et al., 2003; Lobell et al., 2003). The second data source is flux tower eddy covariance (EC) data which provides high resolution $CO_2$ fluxes at point locations (Baldocchi and Wilson, 2001). Previously, data assimilation methods have been used for an ecosystem model in croplands with earth observation data (Revill et al., 2013; Sus et al., 2013), but both studies focused on ecosystem carbon fluxes and leaf area index and included no estimates of yield.

Sites where intensive data collection has taken place do exist and can be very useful in exploring certain aspects of crop physiology, for example in the context of the agricultural model intercomparison and improvement project, AgMIP (Rosenzweig et al., 2013). However, here we aim to explore a general model-data integration system that could be applied to generic farm locations with generally available data. This makes the problem more difficult but the conclusions can be more useful to a general application of the concepts.

In this paper we present a newly developed general, non-crop specific process based model and use parameter inference to infer the most likely parameters for 15 locations for winter wheat and maize using a combination of space-based vegetation indices, eddy covariance flux data and reported agricultural yields. We aim to answer the following questions:

1. Does our model with data constrained parameters predict empirical data better than a model with prior parameters?

2. Are the data constrained parameters similar among different sites and what are the impacts on model predictive accuracy of having site-specific versus site-shared parameters?

3. To what extent does the inclusion of the different types of data in the model fitting process influence the uncertainty in the inferred parameters and model predictions?

We expect the qualitative answer to the first question to be that utilising empirical data does enable the model to make better predictions because that's a typical outcome of our parameter estimation approach. However we are more interested in the

quantitative answer; i.e. how much? For example, the generation of a model that could make extremely precise and accurate predictions would suggest that data-constraining general models with the datasets we identify could provide an extremely useful tools for agricultural predictions and forecasts. Alternatively, the generation of a model that makes very imprecise predictions would suggest that more data collection and model improvement is needed for the model to have practical applications.

In addition to our aims above, our goal with this paper is to provide a proof of concept data-constrained process-based crop model that could be of use in practical agricultural systems. To this end we include more description of the methods than otherwise necessary as well as a more broad discussion of the paper's applicability.

While our study is part of a boarder scientific objective to enable more accurate field scale predictions, the lack of availability of field scale datasets to train and validate our model means that the scale of model evaluation for our study here is a mix of field (flux tower) and regional scales (county and country level for yield estimates and 3 by 3 km scale for photosynthetic activity).

## 2 Datasets used

### 2.1 Study sites

Our analysis focusses on 15 sites for which we can obtain the combination of eddy-covariance data, satellite data and crop yield data for specific crops (summarised in Table 1), of which 7 sites were growing maize (Zea mays) and 8 sites were growing winter wheat (Triticum aestivum; we refer to this simply as wheat). Most of these sites grow maize or wheat on a rotation with other crops and we identify the time period over which the species of interest is growing from the metadata associated with the eddy-covariance data. All of the maize sites are based in the United States. All but one of the wheat sites are based in western Europe, with one site in the United States. For the site where information was available, the crops were not irrigated with the exception of the US-Me1 site (Suyker et al., 2004). All sites have been tilled to a certain degree, generally in accord with agricultural practices in the area. European sites have received a moderate amount of fertiliser (Moors et al., 2010).

### 2.2 Space-based vegetation indices

We use data on vegetation greenness from the MODIS (Moderate Resolution Imaging Spectroradiometer) Terra instrument. MODIS fraction of absorbed photosynthetically active radiation (fAPAR data) from the MOD15A product was downloaded (https://lpdaac.usgs.gov/) for geographic regions corresponding to each of the study sites (Table 1) for the period 2000-2010. This data was subsequently filtered using the quality assurance (QA) indices provided so that only data points calculated using the main algorithm were retained and pixels classified as cultivated land were identified using the MODIS landcover product (MOD12A) IGBP classification.

Using the pixel closest to the flux tower site was infeasible because of data noise and gaps resulting in an uneven timeseries. Instead, we aggregated all pixels within a 3 km by 3 km square centred on the tower site in a single timeseries. The untested assumption behind this aggregation is that farming practices are constant across this scale. To separate between different crops

we use a crop phenology approach (Wardlow et al., 2007). Pixels that a reach maximum fAPAR before day 150 are classed as winter crops (specifically, winter wheat), while crops that peak after that date are classified as summer crops. This procedure is applied for individual years to account for crop rotations.

## 2.3 Eddy-covariance data

We use eddy covariance data for 15 sites across Europe and the United States (Table 1), consisting of 19 data years. The data was obtained from the Ameriflux (http://ameriflux.lbl.gov/) and the European Fluxes Database Cluster (http://www.europe-fluxdata.eu/). We use level 4 data of $CO_2$ fluxes partitioned into gross primary productivity (GPP) and gap filled using the mDS method (Reichstein et al., 2005). Sites that have a crop rotation were filtered to obtain single species timeseries. These include the maize-soybean rotation sites and European mix rotation sites that include winter wheat.

## 2.4 Crop yield data and agricultural dates

To obtain information on crop yield we use data provided by the US Department for Agriculture (USDA) yearly, at the county level, available for the entire study period (https://www.nass.usda.gov/). For the European sites we used country level data provided by the EC Eurostat database, available from 2004 onwards (http://ec.europa.eu/eurostat).

Sowing and harvest dates are required as model inputs and were extracted from the crop calendar global dataset (Sacks et al., 2010). We chose this rather than local level dates for greater model generality.

Fertilizer input data were obtained from the published site descriptions (see Table 1 for references) or from the Nitrogen Fertilizer Application database ((Potter et al., 2010). The model implemented in this study does not require any additional information on irrigation or soil properties.

## 2.5 Environmental input data

We use NASA's Modern-Era Retrospective Analysis for Research and Application (MERRA) dataset (Rienecker et al., 2011) at a spatial resolution of 0.5 degrees latitude by 0.66 degrees longitude and a temporal resolution of 3 hours which we average to a day. Temperature and direct and diffuse photosynthetically active radiation (PAR) data were extracted for each site. Comparison with tower based meteorological data has shown this to be an accurate estimation of conditions at the tower site for all variables and we use MERRA data for the greater generality of the model as this would allow the model to be applied at any location on the globe.

## 3 Model description

Our new general model of crop growth is based on the single plant model of Guilbaud et al. (2014) and, like that model, assumes that annual plants show optimal biomass allocation during vegetative growth and optimal flowering in order to achieve maximum reproductive mass given available resources. Plant growth is divided into three stages, starting at sowing date and ending at harvest: germination, vegetative growth and reproductive growth.

### 3.1 Germination

The germination process is described as a degree day function with a fixed base temperature of $0°C$ up to a parameter germination limit $germ_{lim}$. The accumulated degree days, $germ_{acc}$ are calculated as follows:

$$germ_{acc}(t) = \begin{cases} germ_{acc}(t-1) + (T(t) - T_{base}) & ,T(t) \geq T_{base} \\ germ_{acc}(t-1) & ,T(t) < T_{base} \end{cases} \tag{1}$$

Vegetative growth begins once the accumulated degree days are higher than the limit parameter, $germ_{lim}$, which is a free fitted parameter. initial seed mass is prescribed and is expressed as grams per metre squared, incorporating information about both seed size and planting density. When the germination limit is reached, all seed mass is allocated to above- and below-ground pools according to the optimality criteria described below. Initial model runs have shown that for values of the germination base temperature $T_b$ and seed mass within realistic ranges, the model is largely insensitive to the values of these parameters,

which is why they have been fixed.

### 3.2 Vegetative growth

During vegetative growth, biomass is allocated to either above or below-ground fractions to achieve an optimal carbon to nitrogen (C:N) ratio at the plant level ($\rho$). The net daily growth is calculated as the minimum of a nitrogen limited growth rate, $G_{root}$ and a carbon limited growth rate $G_{leaf}$.

Nitrogen limited growth is considered to be a function of root mass $M_{root}$ and available soil nitrogen $N$:

$$G_{root}(t) = \theta N(t) M_{root}(t-1) \rho, \tag{2}$$

where $\theta$ is the nitrogen uptake capacity of the roots expressed as gN g$^{-1}$ soil N g$^{-1}$ root C day$^{-1}$, $N(t)$ is soil nitrogen at time $t$ (g) and $M_{root}(t-1)$ is the root mass (g) at the previous timestep. Carbon limited growth is considered to be equal to potential net carbon uptake, calculated as the difference between whole canopy photosynthesis and respiration. Photosynthesis

is calculated using the model for C3 plants developed by Farquhar et al. (1980) as described in dePury and Farquhar (1997) and the alternative model for C4 species (Collatz et al., 1992; Von Caemmerer, 2000):

$$G_{leaf}(t) = f(V_{cmax25}, J_{m25}, T(t), I(t), pCO_2, LAI(t-1)) - R_{plant} \tag{3}$$

Here $V_{cmax25}$ is a parameter representing photosynthetic Rubisco capacity ($\mu$mol m$^{-2}$ s$^{-1}$), $J_m25$ is potential electron transport rate and $T$, $I$ and $pCO_2$ are environmental inputs (temperature, solar radiation and atmospheric $CO_2$ partial pressure

respectively). The electron transport rate $J_{m25}$ is represented for fitting purposes as the ratio $fJ$ between $J_{m25}$ and $V_{cmax25}$ to partially eliminate model equifinality. Total absorbed solar radiation $I$ is calculated for direct and diffuse photosynthetically active radiation (PAR) using a sun-shade model (dePury and Farquhar, 1997). Partial pressure of $CO_2$ inside the leaf is calculated assuming a constant optimal ratio $\lambda$ between internal and atmospheric $CO_2$ in the absence of water stress (Haxeltine and Prentice, 1996). See Appendix B for details of the photosynthesis model in equation 3. Leaf area index (LAI)

is calculated from leaf mass $M_{leaf}$ using the leaf mass per area ($LMA$) parameter. Whole plant respiration is calculated as a linear function of total plant mass:

$$R_{plant} = r_{tot}(M_{leaf} + M_{root}) \tag{4}$$

Here $r_{tot}$ represents average respiration per unit plant mass (g g$^{-1}$ day$^{-1}$). This total respiration component accounts for growth costs and maintenance including active nutrient uptake by the roots and is a function of temperature. Given the optimal whole plant C:N ratio that drives the vegetative biomass allocation, this formulation is ultimately equivalent to the nitrogen dependent function commonly used in vegetation models without the need to introduce further parameters for root and leaf specific C:N ratios.

Actual biomass growth is then the minimum between nitrogen and carbon limited growth:

$$G_{net} = min(G_{root}, G_{leaf}) \tag{5}$$

This biomass is allocated to the limiting fraction, either aboveground or belowground in order to adjust the C:N supply. Crops are considered to be not water limited, as all sites are in areas with a high annual precipitation. We lacked any information on soil water availability and initial trials to data-constrain a model that included the effects of varying soil water availability led to poorly constrained parameters related to soil water constraints (see section 7).

## 3.3 Optimal flowering and reproductive growth

Reproductive growth starts at a point where the supply of any of the resources, carbon or nitrogen, reaches a maximum, which we term 'peak resource'. This is the point in time which will result in the maximum final reproductive mass as further increase in vegetative fractions would not result in an overall increase in growth rate and lead to suboptimal growth (see Guilbaud et al. (2014) for an in depth discussion of this).

The peak nitrogen condition is achieved when an increase in root mass does not result in an increase in nitrogen uptake. This condition is achieved in nitrogen limited environments where the nitrogen available in the soil is depleted through the period of vegetative growth. This assumption can be considered valid in agricultural systems where the major nitrogen input into the system during the growing period comes solely from agricultural fertilisers. Soil nitrogen decays monotonically through the season in our model due to the simplicity with which we model nitrogen uptake and so detecting the peak nitrogen condition is straightforward. Similarly, the peak carbon flowering condition is triggered when the addition of aboveground biomass would not lead to an increase in net carbon gain, due to self-shading in the canopy. To calculate the peak carbon trigger we use the environmental variables averaged over $p$ days, to avoid flowering being triggered by short-term environmental fluctuations. We infer $p$ alongside the other parameters in our model.

During the reproductive phase all new biomass produced is assigned to reproductive tissues. Nitrogen and carbon are translocated to reproductive organs at a constant rate, $m_{trans}$. As all biomass within the model is calculated as mass of carbon and agricultural yield data is reported as total dry mass we use a conversion parameter to account for the carbon fraction, $C_{frac}$. This parameter also accounts for the differences in total reproductive mass and actual mass harvested and reported as yield.

## 4 Parameter estimation technique

We use Bayesian parameter inference techniques to infer the parameters for the model described above. The technique involves solving Bayes' theorem which in this context states

$$P(\theta|obs) = \frac{P(obs|\theta)P(\theta)}{\int P(obs|\theta)P(\theta)d\theta}, \tag{6}$$

where $P$ denotes a probability, $obs$ is the empirical data, and $\theta$ is the set of parameters to be inferred (Gilks, 1996.). The term in the denominator can be treated as a normalising constant in our study and so we omit it here. Thus our problem reduces to $P(\theta|obs) \approx P(obs|\theta)P(\theta)$ where $P(obs|\theta)$ is usually referred to as the likelihood of the data given the model and $P(\theta)$ is the prior probability of the parameters. Prior probabilities of parameters can be determined by previous empirical evidence, such as field measurements. In our case we do not have any prior expectations about what the prior parameter values should

be and so we specify that each parameter is equally likely to fall within a wide range of values (flat priors). This means that our study reduces to inferring the joint probability distribution of the parameters based on the likelihood of the data given all possible parameter combinations. We cannot solve this inference problem exactly. Instead we use Markov-Chain Monte Carlo techniques with the Metropolis Hastings algorithm to approximate the likelihood and its associated joint parameter probability distribution, which we implemented using the Filzbach inference library as detailed in (Caldararu et al., 2012). This algorithm

works by iteratively making random mutations to an existing parameter set, computing the likelihood associated with the new set of parameters, and then replacing the existing parameter set with the new set based on the ratio of their likelihoods according to the Metropolis-Hastings algorithm (Gilks, 1996.). Parameter ranges were set based on literature and our understanding of plausible biological ranges for these crop species and agricultural scenarios as well as additional adjustment to ensure parameter convergence during inference.

Three different datasets were used in combination to infer our model parameters - MODIS fAPAR, flux tower GPP and crop yield data. Each dataset contributes to the assessment of the model likelihood but each one of these has different temporal resolutions and covers different time periods, resulting in a variable number of data points. To prevent our inferred parameters from being overly-based towards explaining the datasets with the greatest quantity of data points we down-weighted the contributions to our likelihood estimates from each data point according to the quantity of data in each data set. The likelihood

function used in Filzbach is therefore:

$$l(Z_{\mathbf{x}}|\theta_{\mathbf{x}}) = \sum_D \frac{1}{N_{\mathbf{x},\mathbf{D}}} \sum_{t(\mathbf{x},\mathbf{D})} \ln[n(Y_{obs}(\mathbf{x},D,t), Y_{pred}(\mathbf{x},D,t,\theta_{\mathbf{x}}), \sigma_{\mathbf{x},\mathbf{D}})], \tag{7}$$

where $\theta_{\mathbf{x}}$ is the vector of model parameters at site $\mathbf{x}$, $N_{\mathbf{x}}$ is the number of data points in each dataset $D$ at each location and $n(Y_{obs}(\mathbf{x},D,t), Y_{pred}(\mathbf{x},D,t,\theta_{\mathbf{x}}), \sigma_{\mathbf{x}})$ denotes the probability density for observing $Y_{obs}(\mathbf{x},D,t)$ given a normal distribution with mean $Y_{pred}(\mathbf{x},D,t,\theta_{\mathbf{x}})$ and standard deviation $\sigma_{\mathbf{x},\mathbf{D}}$ which expresses the magnitude of unexplained variation in the

variable $Y$. $Y$ refers to the model variables corresponding to the three datasets. Note that with this definition of the likelihood

we are treating every data point as independent, that is the likelihood of a value at time $t$ is treated independently from the likelihoods at preceding times. This is only an approximation but is commonly used in parameter estimation studies because the additional mathematical and computational complexity of accounting for non-independent data.

We adopt different techniques to estimate the standard deviation $\sigma_{\mathbf{x},\mathbf{D}}$ above, depending on the dataset $D$ at each location. Generally, we assume that the variation in the model predictions about the data is solely due to uncertainty in the data. We address the limitations of this assumption and future improvements in the Discussion. The GPP data do not have an estimate of uncertainty and so we infer the uncertainty associated with those data as the parameter $\sigma_{\mathbf{x},\mathbf{D}}$. In the case of MODIS fAPAR data we explicitly incorporate a measure of variation in the data within the geographical area used to compute the mean fAPAR as well as inferring a parameter representing additional unexplained variation. We include this parameter to account for known issue in space based remotely sensed data, such as background soil reflectance. The crop yield data already have estimates of observational uncertainty associated with them and so we use those data to define $\sigma_{\mathbf{x},\mathbf{D}}$.

## 5  Experimental protocol

In order to assess whether the model with data constrained parameters predicts empirical data better than a model with prior parameters we infer the parameters for each site individually using all of the empirical data and compare the model predictive performance to one in which the parameter values are sampled randomly from the prior range.

We compare the inferred parameters and predictive performance of models with parameters inferred using data from individual sites (the one site model) or from multiple sites together (all sites model), always keeping maize and winter wheat sites separate, to assess the effects of allowing parameters to differ between the sites. Preliminary investigations revealed that similar model parameter distributions were inferred once data from more than 3 sites were used in combination when inferring the parameters. We therefore also take the opportunity to assess the performance of the models with parameters shared between sites in predicting data that has not been used in parameter inference (evaluation model).

To assess the importance of different types of data-constraints we perform a data knock out experiment and we infer the model parameters for individual sites using only one or two of the different empirical datasets and assess inferred model parameters and model performance.

In general we assess model predictive performance by quantifying the root mean squared error between the model predictions and the empirical data to access model precision and the mean error to assess model bias. We normalise both these metrics by the mean value of the different empirical dataset types to aid in comparison. We calculate parameter uncertainty as the 95th percentile confidence interval from the posterior distribution (Section 4).

To calculate uncertainty for the model predictions we sample parameter values from their respective posterior distribution and compute predictions with each parameter combination, which results in a corresponding distribution of model predictions. We report this prediction distribution uncertainty using 95th percent confidence intervals. This predicted distribution does not include the prescribed or inferred uncertainty about observations, $\sigma_{\mathbf{x},\mathbf{D}}$, our predicted distributions correspond to the state being predicted and not the observations of that state.

# 6 Results

## 6.1 Prior and posterior model predictions

In general and as expected, the predictive accuracy of both the wheat and maize models is improved by inferring their parameters; the root mean squared error and bias of the model predictions is reduced for predicting all empirical datasets compared to the prior model (Table 3). These improvements are about a 40% reduction in RMSE for both GPP and fAPAR and an 80% reduction in RMSE for yield. Visual inspection of the predicted time series for the models with prior and posterior parameter distributions (e.g. Figure 1 for wheat in one site) highlights that the model with prior parameters predicts the same qualitative behaviour as the model with inferred parameters but that parameter inference reduces the posterior uncertainty in the predictions of the model.

In terms of uncertainty, the posterior models show a large reduction when compared to the prior of aboveground biomass (86%) and yield (97%), but a smaller reduction for the belowground variables (67% for root biomass and 20% for soil nitrogen), as there is no data in the fitting procedure to directly constrain these. Visual inspection also emphasises the importance of model structural constraints on the model dynamics e.g. the model predicts a narrow range of dynamics in some properties at certain times of the year (e.g. biomass in leaves, roots and reproductive parts soon after sowing) irrespective of the parameter values.

## 6.2 One site vs. all sites fit

On average the RMSEs are very similar between the models with parameters inferred for individual sites to when parameters are inferred for all sites together (Table 3). In general, we expect that if we were to infer a single set of parameters for individual sites then the predictive performance of that model will always be at least as good as when the set of parameters has been inferred for all sites. This may not necessarily be the case when inferring parameter probability distributions: the lower quantity of data could result in greater parameter uncertainty which may on average lead to a lower predictive accuracy than that using the more constrained parameter distributions obtained by inferring parameters from all sites. This explains why some of the mean RMSE scores are higher for the model with parameters inferred from individual sites. The bias scores are also very similar although the bias tends to be smaller on average for the models with parameters inferred using all sites.

As expected, the uncertainty in the predicted GPP, fAPAR and yield is lower for the models with parameters inferred using all sites because more data is used to infer the parameter values for those models, leading to lower uncertainty in the inferred parameter distributions (Figure 2). When parameters are inferred for individual sites uncertainty is around 134% for GPP, 121% for fAPAR and 33% for yield, with similar values at wheat sites (Table 3). This is reduced to around 45% for GPP, 100% for fAPAR and 12% for yield estimates when parameters are inferred using data for all sites. Visual inspection of the change in uncertainty over time highlights that prediction uncertainty due to parameter uncertainty is highest at the start and end of the season (over 100%) but decreases to 50% on average for all variables in the middle of the growing period (Fig. 4).

Inspection of the inferred parameter distributions (Fig. 2) shows, as expected, that the posterior parameter uncertainty tends to be higher when parameters are inferred using data from individual sites versus using all sites together, although these distributions overlap for almost every site and every parameter. In general, these inferred parameter distributions show greater

differences between winter wheat crops and maize crops than they do as a result of using more sites for inference. One exception is the sole winter wheat site in the United States; inferred to have lower soil nitrogen, respiration rate and translocation rate of mass from vegetative to reproductive tissue. These inferred differences are probably due to differences in winter wheat crops between the USA site and the European sites such as different crop varieties or agronomic practices.

Visual inspection of the predicted time series of GPP, fAPAR and yield for maize and winter wheat predominantly show very similar predictions between the models with parameters estimated from one site versus all sites (Figure 3 shows predictions for representative sites. Appendix A shows timeseries for all sites with associated uncertainty). There tends to be greater differences between the model predictions and the empirical data when the model has site-specific parametrisations than when parametrisations are shared between sites. The one notable exception is again the winter wheat site in the US, for which inferring parameters for the specific site leads to much more accurate predictions compared to the model with parameters inferred for all sites (Fig. A1). Other than that site, the time series for GPP, fAPAR and yield for maize show larger discrepancies between the data and the model predictions than from the predictions of different models. GPP tends to be reproduced well, relative to the other time series, with an average correlation coefficient of around $r^2$=0.7. fAPAR is predicted less well (around $r^2$=0.4) which is at least partly due to a systematic under prediction of fAPAR at the start and end of the year. We attribute this to the fact that the fAPAR data reflects the light absorption by plants in a region that includes vegetated areas out with just the fields whereas the model is predicting only light absorption by the crop (discussed further below). Annual yields are predicted least well by our models (around $r^2$=0.1) and we attribute this at least in part to the data itself having a relatively high uncertainty (discussed further below).

We evaluate the model transferability by inferring the model parameters using a subset of the sites and assessing model predictive performance against the remaining sites (Fig. 3 and Table 3). In general, the model RMSE and bias do not differ between the sites that were used for parameter estimation and those that were not. Moreover, the model predictive performance is similar to that resulting when fitting to all sites. The uncertainty for GPP, fAPAR and yield at maize sites is similar to that obtained by fitting to all sites, but for the wheat sites the uncertainty in GPP and fAPAR increases, while the yield uncertainty remains at the level obtained when fitting to all sites (Table 3).

## 6.3   Impacts of using different data types

Our data type hold out experiments show clear differences in the roles played by different data types in improving model predictive accuracy, but the effects are similar for both crop types (Figure 5, this figure only shows model RMSE and bias when parameters are inferred using data for individual sites but the results are similar when all sites are used to infer model parameters). The largest effect of adding a given data type is when yield data is included, which significantly reduces RMSE and bias for predicting yield. This makes intuitive sense, although interestingly including yield data alone and as part of a combination also tends to improve model predictive performance for GPP and fAPAR. Counter intuitively, including GPP data alone or fAPAR data alone only has subtle effects on the model RMSE and bias for predicting those variables and yield, but including those datasets in combination does indeed lead to improvements in RMSE and bias.

The greatest improvements in model predictive performance for all response variables is obtained when all data types are used for parameter inference. This is not inevitable as an overall more likely model might be achieved by sacrificing predictive accuracy for one data type in order to improve predictive accuracy for another. For example, adding fAPAR data alone slightly improves model RMSE for fAPAR data, but makes it worse for GPP and yield predictions when compared to the model with prior parameter distributions. Indeed the crops do not flower for maize or wheat when only fAPAR data is used for parameter inference. Comparing knockouts with and without fAPAR data included implies a trade-off between predicting the fAPAR data well and predicting GPP well (Figure 5). Interestingly, all models underestimate GPP, although this bias is least when all data is used to infer the model parameters.

The uncertainty in model predictions (Figure 6) follows a similar pattern to model error, with the fAPAR only model having the highest uncertainty (up to 900% for GPP) while the GPP and fAPAR model performs best with uncertainty values of 123%, 128% and 32% for GPP, fAPAR and yield respectively, values which are close to those obtained through fitting to all the data. The GPP and yield model also has relatively low uncertainty values for GPP and fAPAR estimates but fails to produce any yield at the wheat sites (the plants do not proceed to the flowering stage).

## 7  Discussion

### 7.1  Model performance

We show that a process-based crop model constrained using EC data, satellite fAPAR observations and regional yield estimates can improve model performance compared to the model run with prior parameter ranges and greatly reduces the uncertainty in model output. However, the resulting uncertainty in both state variables and model parameters is still relatively high.

Model uncertainty is difficult to compare with previous crop modelling studies, as models with fixed parameter values do not often provide uncertainty estimates. In fact, providing uncertainty values for all model variables and parameters is one of the advantages of a data constrained model. In the current model, uncertainty is highest at the start of the season for all variables but decreases rapidly and final yield uncertainty is much lower. This is due to thresholds: abrupt changes from one growing stage to another when small differences in parameters can lead to large differences in resulting variables. It is, however, important to note that the uncertainty in our yield predictions remains high and the model in its current form is unlikely to provide accurate predictions for practical applications without the addition of new data (Section 7.4). We have however shown that the use of three different data types does reduce prediction uncertainty - pointing to an avenue for future model improvement.

Our estimates of model parameter uncertainty, and consequently model prediction uncertainty, are influenced by our assumption that the model is correct and that any departure of the data from predictions is due to measurement error. This is undoubtedly false but makes our parameter estimation method simpler. Overall prediction uncertainty can be decomposed into initial condition uncertainty, parameter uncertainty and model uncertainty and methods exist for making these uncertainty estimates and building them into predictions (Wallach et al., 2016b, a). Such estimates should be made if our model is applied to real agricultural prediction scenarios.

In terms of the posterior parameter distributions, resulting parameters show a similar degree of constraining to that observed in previous model parametrisation studies for natural ecosystems (Keenan et al., 2012). The photosynthesis related parameters are badly constrained despite the fact that GPP estimates have a relatively low uncertainty. This can be explained by the structure of the photosynthesis component which is rigid compared to other components of the model as these processes are better understood. In contrast, belowground processes are both poorly understood and lack the data to properly constrain model parameters (Pendall et al., 2004).

In terms of model performance, the model correctly predicts seasonal trajectories of GPP and final yield data. We cannot however capture the interannual variability in yields, which is most likely due to the fact that our model does not include a response to water limitation or heat damage. The fact that we use regional yield data can also lead to discrepancies between the yield at each specific flux tower site and the yield data. The model does not capture the fAPAR seasonal cycle well, especially at the maize sites, which is due to the low spatial resolution of the data. However, the predicted model fAPAR is more realistic than the fAPAR data, which is one of the advantages of using a process-based model with a more rigid structure than a statistical one.

One additional complication is the different spatial scales of the three datasets - while the eddy covariance data is at the scale of the flux tower footprint, which can be seen as equivalent to the individual field scale, the fAPAR and yield data correspond to larger scales (county and country level for the yield data and a 3 by 3 km scale for the fAPAR data). The assumption behind our analysis is that the conditions at field scale are representative of the regional scale, so that there would be no discrepancy between model predictions at these different scales. This is obviously a source of error, especially at the wheat sites in Europe, which will be located over a much more heterogeneous landscape. Further sources of data at the field scale would be required to identify the model error caused by the discrepancy in spatial scales.

## 7.2 Use of the different datasets

Eddy covariance data is to date the most widely used data set for parametrisation of vegetation models (Fox et al., 2009; Xiao et al., 2011). We show that removing this data from the fitting procedure does not radically decrease model performance. If we consider what information content this data provides - primary productivity and $CO_2$ flux seasonality, this fact is maybe not surprising. The seasonality information is already contained in the fAPAR dataset, while the primary productivity is highly constrained by the structure of the biochemical photosynthesis model. Furthermore, the GPP only fit results in an underestimation of the final yield, indicating that the sole use of EC data in crop models is not sufficient to accurately predict yields. Unlike most studies using EC data we have used sites with only one year of data as these were the only available agricultural sites and it is possible that more GPP data at one site could increase its importance in the fitting. EC data could also be a valuable tool for independent model evaluation as it provides information about plant function not included in the other available data.

Space based vegetation data has the main advantage of a large spatial and temporal coverage, so that it can be used irrespective of the local monitoring infrastructure, providing a general data source. However, the quality of the data is relatively low, especially at the high spatial resolutions needed for crop modelling. This problem is particularly obvious in the case of the maize data, which lacks the expected seasonality and is reflected in the very high error in the fAPAR only fit. However, the

model fits without fAPAR (GPP and yield only) show a high error as well, indicating that the information content in vegetation indices is needed for constraining the model but not sufficient.

Some of these limitations are not general for remote sensed data, but can be attributed to the spatial and spectral resolution of the MODIS instrument. The 1 km spatial resolution can be too coarse for agricultural fields, especially in areas with heterogeneous landcover. Other existing instruments, specifically the Landsat family, have a better spatial resolution (30 m), but a much poorer temporal resolution which we have found unsuitable for fitting a plant growth model where developmental changes can be abrupt. More recent missions, such as Sentinel-2 will have more suitable spatial and temporal resolutions for use with this type of model (Herrmann et al., 2011). Some of the error in the data can also be attributed to misclassification of pixels. We use a simple phenology based approach which is one of the only ones available for data with a relatively wide bandwidth such as MODIS. This method is useful for winter crops which have different timing compared to the natural vegetation, but less useful for summer crops such as maize where there is no clear separation in phenology between cropland and the surrounding vegetation. Hyperspectral data can be used more accurately for crop identification (Thenkabail, 2001) but to date no space-based instrument is available that has the required bandwidth, the spatial and temporal coverage and the spatial and temporal resolution. However, such data should be used at local scales if the measurements are available.

Crop yield is the data that is traditionally used for evaluating agricultural models and is arguably the most important to predict correctly, given that the purpose of the model is to predict crop productivity. We have used county and country level reported yields rather than field level measured yield because of both the availability of the data and the generality of the method. The model fitted with yield data only gives a good fit to yields, but higher errors for the GPP and fAPAR estimates which raises questions about the correctness of models which only use final yields to assess performance and the ability of such models to predict crop yields under different conditions. Crop yield data provides the final point of plant crop growth but there is potentially a multitude of model structures and parameter combinations that can result in that yield.

In addition to the three datasets used for parametrisation, the model also requires input data in the form of sowing and harvest dates and fertiliser inputs. Additional uncertainty is associated with these datasets which is not available nor accounted for in our analyses. For example, the crop calendar (Sacks et al., 2010) and Nitrogen Fertilizer Application ((Potter et al., 2010)) datasets are global data collections that will imperfectly represent the value for any given location. Alternatives to these global datasets would be to use location-specific data, or to infer the values. Location specific data has the advantage of more accurately reflecting the situation at a given site and would therefore be useful when the model is applied at the field scale, but such data is unlikely to be available for all sites. Successful inference of the values would depend on if there is enough information in the datasets used to infer the model parameters. If there is inadequate data then there would be excessive degrees of freedom for inference, leading to the wrong parameter values being inferred and the model performing poorly in novel situations. Therefore the decision whether to obtain more data or infer unknown quantities in future applications of our model and inference framework depends on the data availability and the intended scales of application.

## 7.3 Choice of model

Here we have chosen a given model structure and extensively tested the way in which constraining the parameters with different datasets in different configurations. The question that arises is to what extent the chosen model itself affects the present results. We have chosen a novel, physiology based model which includes plant optimality concepts, which on one hand has the advantage that it is more general than some of the older models and lacks artificially set thresholds between growth stages, but does have the disadvantage of being less thoroughly tested against field observations. An ideal companion paper to this study would be a comparison of different model structures with a constant data constraining framework, providing greater insights into which parts of the model lead to high errors or uncertainty. However, given the limitations of the current study, we acknowledge this limitation and report most error metrics as relative to prior model runs in an attempt to isolate errors created by the data and model fitting from those caused by the model itself.

## 7.4 Future data needs

The fact that our model shows a relatively good fit when constrained at multiple sites indicates that it would be possible to obtain a single parameter set for one cultivar given the same agricultural practices, so that the model can be fitted at a small number of locations and then applied more widely. However, the parameters are badly constrained and part of the data we have used is not sufficiently accurate to allow the use of the model at a wider variety of locations and climate conditions. Accurate yield data is essential but not sufficient and must be accompanied by a growth timeseries. Our results indicate that additional EC data is not necessary, especially given the cost of installing and maintaining a flux tower. Instead, either biomass or LAI (or fAPAR or other VIs) data could be easier to obtain at multiple locations. The belowground part of the model, describing root nitrogen uptake, is only indirectly constrained by the existing data and any observation of root mass and function would have the capacity to add extra information, especially timeseries information (Johnson et al., 2001).

The model in the version presented in this paper does not include any water limitation to growth due mainly to a lack of data constraint on any water related parameters, as we found that latent heat data from EC towers is not sufficient. Below-ground measurements of not only root growth but also soil water properties would again provide some of the necessary information.Such belowground data, especially if supplemented by nutrient concentrations can also help constrain a more complex version of the nitrogen uptake scheme, which could be improved to include more explicit soil-plant interactions and additional processes such as biological nitrogen fixation for legumes.

If this model, or any other similar process-based data constrained crop model, is to be used for scientific purposes to understand the response of crops to climate across the globe, the ideal data would be a global data set, such as space-based vegetation observations, combined with high quality field level data that would ideally include growth timeseries, final grain yield and information about agricultural practices. However, if the model is to be used for agricultural purposes, to aid decision making at the local level, high quality field level data would be sufficient. A valuable evaluation in such studies, not conducted here for brevity and due to a lack of location-specific data, would be to compare the predictive accuracy of the model against the

predictive accuracy of a statistical average over the data. Such an analysis would reveal whether and how much benefit is gained by using a data constrained model for predictions.

## 8 Conclusions

In this paper we present a method for data constraining a process-based agricultural model to three sources of data: eddy covariance flux measurements, space-based fAPAR and regional yield estimates. We show that the data constrained model performs better than the model with prior parameter estimates, especially in terms of uncertainty and even though the data used is in some cases not sufficient to fully constrain posterior parameters it has sufficient information value to be used for model parametrisation. We apply the model to both maize and wheat sites and show that the model performs equally well for both species. Parameters can be shared between sites of the same species with a similar performance to local parameters and reduced uncertainty. We have also investigated the impact of the different data sets on constraining the model and we show that all three types of data contribute to the model performance, but that if in a data limited world one of the data types was not available, the model can be constrained reasonably well with fAPAR and yield data only. There are still gaps in the data available for model parametrisation, which are also a limitation to the models which can be parametrised, in particular in relation to water limitation on crops and we believe that a model parametrisation framework such as that presented here can help identify those gaps and the data needed to further our capacity to model crops.

## 9 Code availability

All model code used in this paper is available from the authors upon request.

## 10 Data availability

All data used in this paper is freely available and has been fully referenced in the text.

## Appendix A: Site level model simulations

Figures A1-A3 show site level predictions for the one site and all site model parametrisation. Figures A4-A6 show results from the site knock out evaluation.

## Appendix B: Photosynthesis model

In the current study we use the standard biochemical model of Farquhar et al. (1980) for C3 photosynthesis, using the parameter values from dePury and Farquhar (1997). The model stipulates that the photosynthesis rate is defined as the minimum of two

rates, Rubisco limited photosynthesis, $A_v$, and electron transport limited photosynthesis:

$$A = min(A_v, A_j). \tag{B1}$$

Rubisco limited photosynthesis is a function of the parameter $V_{cmax25}$, adjusted for temperature and the internal $CO_2$ partial pressure, $c_i$:

$$A_v = V_{cmax} \frac{c_i - \Gamma_*}{c_i + K'} \tag{B2}$$

Here $V_{cmax}$ is the adjusted for temperature value of $V_{cmax25}$ using the Arrhenius function. See table A1 for definitions and values of photosynthetic parameters. The electron transport limited rate is calculated as:

$$A_k = J \frac{c_i - \Gamma_*}{4(c_i + 2\Gamma_*)} \tag{B3}$$

where $J$ is the solution to the quadratic equation:

$$\Theta J^2 - (I_a + J_m)J + I_a J_m = 0 \tag{B4}$$

Here, $J_m$ is the temperature adjusted value of the model parameter $J_{m25}$ and $I_a$ is the photosynthetically active radiation (PAR) absorbed by the photosystem:

$$I_a = I \frac{(1-f)}{2} \tag{B5}$$

The parameters $V_{cmax25}$ and $J_{m25}$ are free parameters in the model (Table 2) and are the values of carboxylation capacity and electron transport at a temperature of $25°C$, while $V_{cmax}$ and $J_n$ are the parameters at the current temperature, calculated using the Arrhenius function.

The internal $CO_2$ partial pressure is calculated based on the assumption that plants maintain a constant ratio between atmospheric and internal partial pressure in the absence of water stress:

$$\lambda = \frac{c_i}{c_a} \tag{B6}$$

where $c_a$ is the atmospheric $CO_2$ partial pressure and is a model input and $\lambda$ is a free model parameter.

In the case of C4 photosynthesis, the standard biochemical model includes a third limitation, the PEP-carboxylation rate (Collatz et al., 1992; Von Caemmerer, 2000) and we have used a simplification of this model, adapted from Haxeltine et al. (1996) which uses different biochemical constants to reach an equivalent photosynthesis rate using only the Rubisco and electron transport limited rates, which is independent of $CO_2$ and temperature in non-extreme conditions.

We calculate the PAR absorbed by the canopy as a sum of absorbed direct and diffuse radiation:

$$I = I_{direct0}(1 - e^{k_{direct}LAI}) + I_{diffuse0}(1 - e^{k_{diffuse}LAI}) \tag{B7}$$

where $k_{direct}$ and $k_{diffuse}$ are light extinction coefficients for the direct and diffuse components of radiation respectively and $I_{direct0}$ and $I_{diffuse0}$ are the two respective components of PAR at the top of the canopy and are environmental drivers for the

model. The diffuse radiation coefficient is assumed to be a constant while the direct extinction coefficient varies with day of year and latitude as follows:

$$k_{direct} = \frac{0.5}{sin\beta} \tag{B8}$$

where $\beta$ is the sun elevation angle:

5 $\quad sin\beta = sin\Lambda sin\delta + cos\Lambda cos\delta \tag{B9}$

Here $\Lambda$ i s the site latitude and $\delta$ is the sun declination angle calculated at noon, given the model timestep of one day, as a function of day of the year, $DOY$:

$$\delta = 23.45sin(2\pi\frac{DOY + 284}{365}) \tag{B10}$$

*Author contributions.* All authors contributed to model development and analysis

10 *Competing interests.* There are no competing interests

*Acknowledgements.* We would like to acknowledge all data providers for the eddy covariance flux sites. Funding for AmeriFlux data resources was provided by the U.S. Department of Energy's Office of Science. We would also like to thank the developers of the MODIS fAPAR product used in this study. We thank Christoph Müller, Daniel Wallach and an anonymous reviewer for their constructive comments that greatly improved our manuscript.

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

**Table 1.** Study sites. All sites correspond to eddy covariance measurement sites

| Site name | Coordinates | Crop | Country | Irrigation | Reference |
|-----------|-------------|------|---------|------------|-----------|
| **Mead 1** | 41.1651,-96.4766 | Maize | United States | Irrigated | Suyker et al. (2004) |
| **Mead 2** | 41.1651,-96.4766 | Maize rotation | United States | Irrigated | Suyker et al. (2004) |
| **Mead3** | 41.1651,-96.4766 | Maize rotation | United States | Rainfed | Suyker et al. (2004) |
| **Bondville** | 40.0062,-88.2904 | Maize rotation | United States | Rainfed | Meyers and Hollinger (2004) |
| **Rosemount 1** | 44.7217,-93.0893 | Maize rotation | United States | N/A | Griffis et al. (2007) |
| **Rosemount 3** | 44.7217,-93.0893 | Maize rotation | United States | N/A | Griffis et al. (2007) |
| **Fermi** | 41.8593,-88.2227 | Maize rotation | United States | N/A | - |
| **ARM Great Plains** | 36.6058,-97.4889 | wheat | United States | N/A | Fischer et al. (2007) |
| **Risbyholm** | 55.5303,12.0972 | Wheat rotation | Denmark | Rainfed | Moors et al. (2010) |
| **Auralde** | 43.5494,1.1078 | Wheat rotation | France | N/A | Moors et al. (2010) |
| **Gebesee** | 51.1001,10.9143 | wheat rotation | Germany | Rainfed | Moors et al. (2010) |
| **Grignon** | 48.844,1.9524 | wheat rotation | France | Rainfed | Moors et al. (2010) |
| **Klingenberg** | 50.8929,13.5225 | wheat rotation | Germany | Rainfed | Moors et al. (2010) |
| **Lonzee** | 50.5522,4.7448 | wheat rotation | Belgium | Rainfed | Moors et al. (2010) |
| **Lutjewad** | 53.3833,6.3667 | wheat rotation | Netherlands | Rainfed | Moors et al. (2010) |

Meyers, T. P., Monson, R. K., Munger, J. W., Oechel, W. C., Paw, U. K. T., Schmid, H. P., Scott, R. L., Starr, G., Suyker, A. E., and Torn, M. S.: Assessing net ecosystem carbon exchange of U.S. terrestrial ecosystems by integrating eddy covariance flux measurements and satellite observations, Agricultural and Forest Meteorology, 151, 60 – 69, doi:http://dx.doi.org/10.1016/j.agrformet.2010.09.002, http://www.sciencedirect.com/science/article/pii/S0168192310002479, 2011.

5   Ziehn, T., Scholze, M., and Knorr, W.: On the capability of Monte Carlo and adjoint inversion techniques to derive posterior parameter uncertainties in terrestrial ecosystem models, Global Biogeochem. Cycles, 26, GB3025–, doi:10.1029/2011GB004185, 2012.

Zwart, S. J. and Bastiaanssen, W. G.: Review of measured crop water productivity values for irrigated wheat, rice, cotton and maize, Agricultural Water Management, 69, 115 – 133, doi:http://dx.doi.org/10.1016/j.agwat.2004.04.007, http://www.sciencedirect.com/science/article/pii/S0378377404001416, 2004.

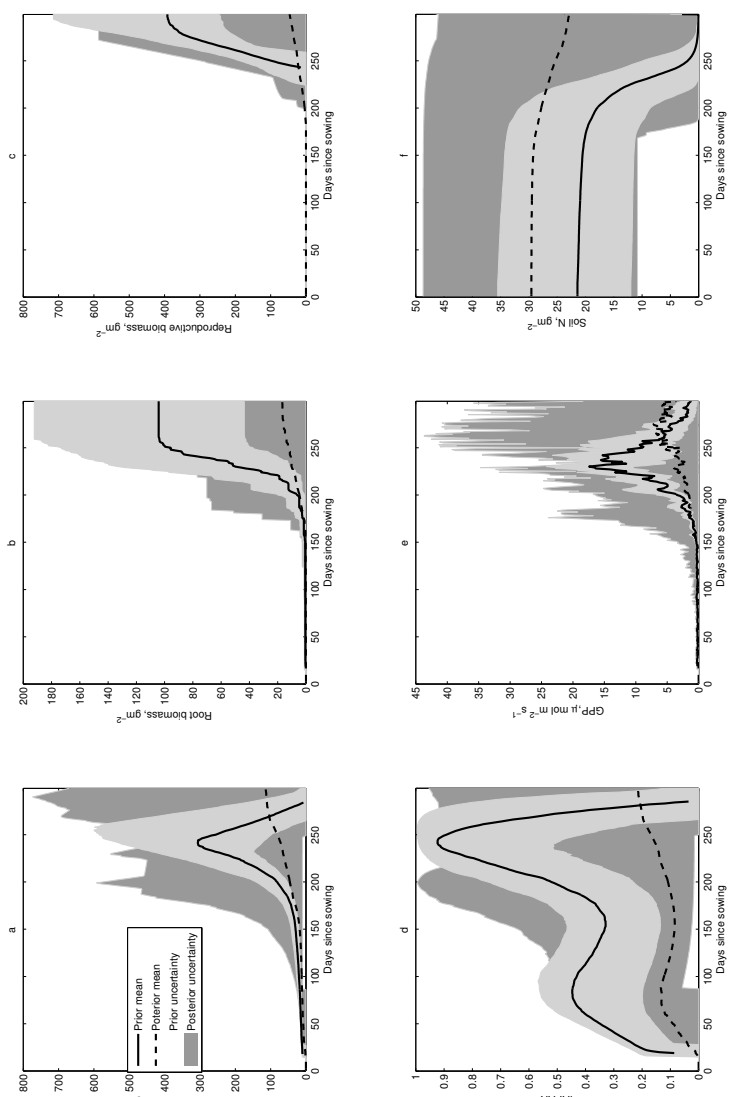

**Figure 1.** Comparison of prior model predictions (dark grey, dashed line) and posterior model predictions (light grey, continuous line) at one wheat (DK-Ris) site. Panels show (a) Aboveground biomass, (b) belowground biomass, (c) Reproductive biomass, (d) fAPAR, (e) GPP and (f) soil nitrogen.

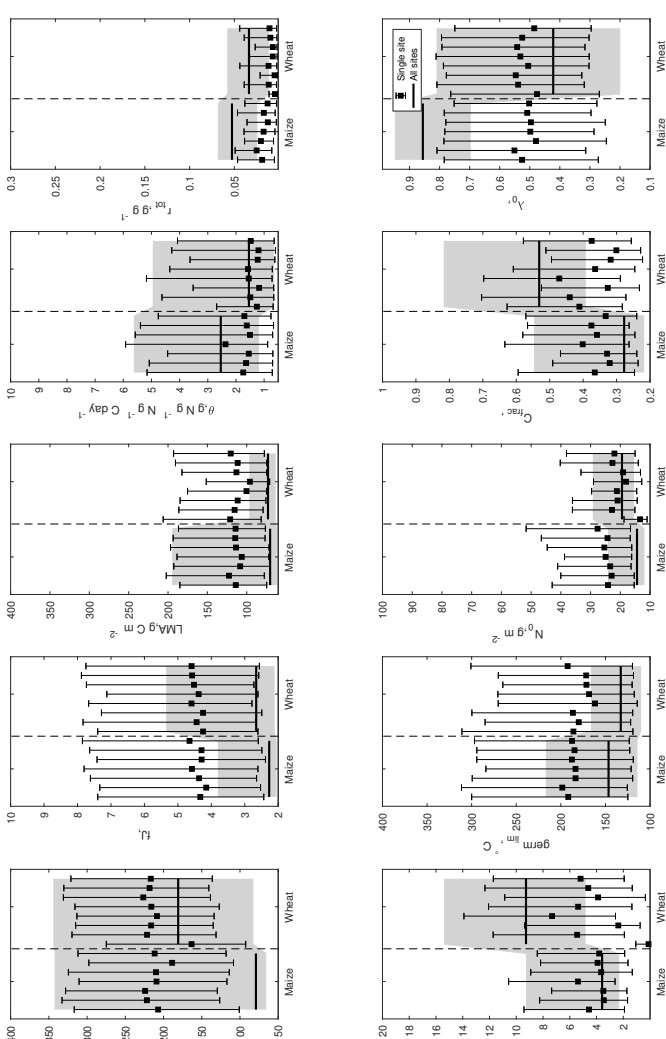

**Figure 2.** Estimated model parameters for all sites, fitted to individual locations (circles) and all locations combined (black line). Values are posterior medians and error bars and shaded areas represent 95th percentiles of the posterior parameter distribution for the one site and all sites parametrisation respectively.

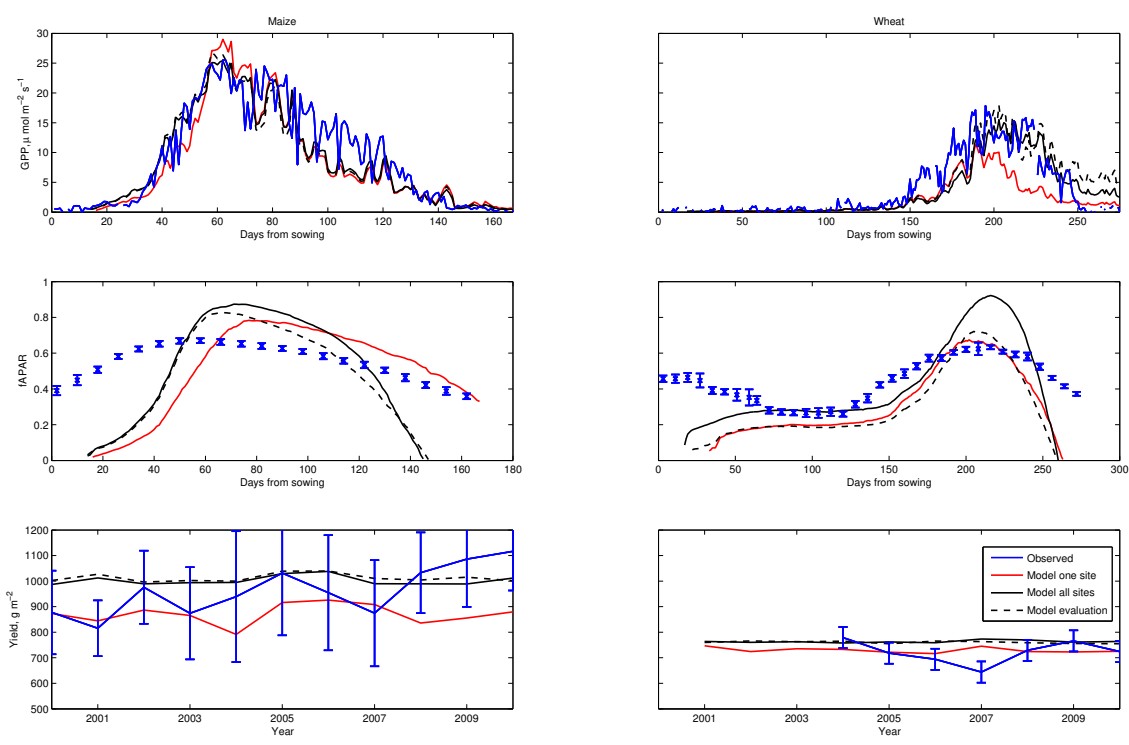

**Figure 3.** GPP, fAPAR and yield model predictions at one maize (US-Ro3) and one wheat(DE-Gri) site. Figure shows posterior mean predictions for the one site, all site and evaluation model fit. Neither site has been included in the evaluation fitting.

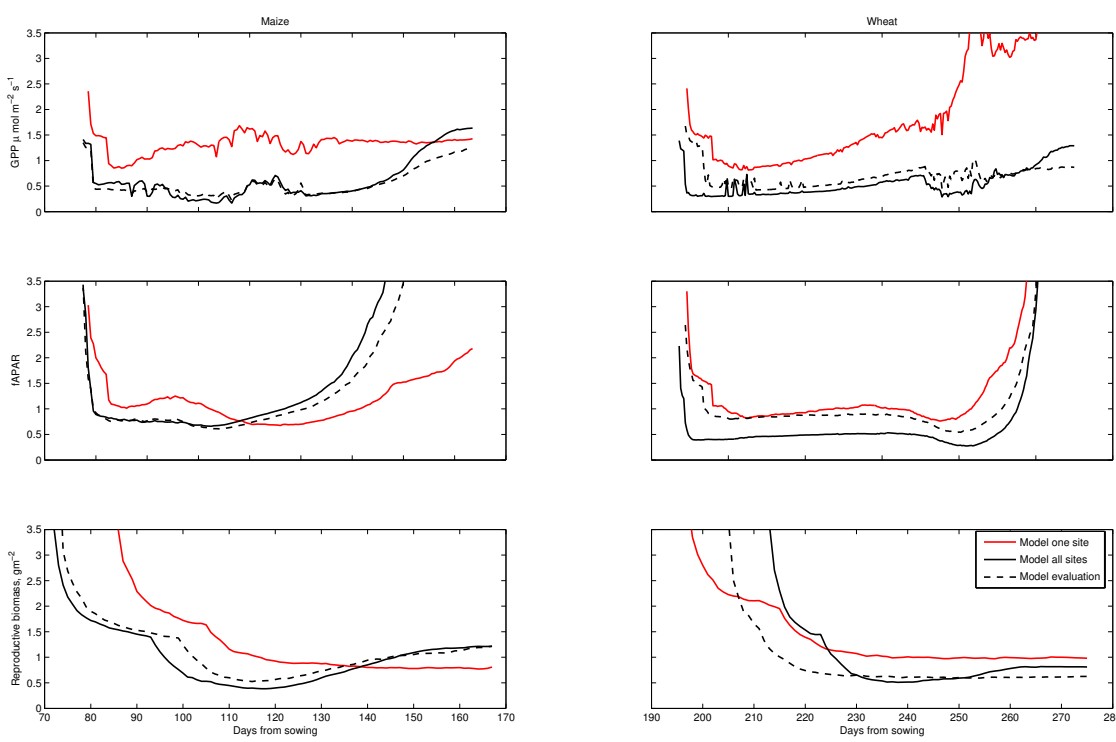

**Figure 4.** Normalised uncertainty for GPP, fAPAR and yield model predictions at one maize (US-Ro3) and one wheat (DE-Gri) site. Uncertainty is calculated as 95th percentile confidence bounds normalised by the posterior mean for the one site, all site and evaluation model fit. Neither site has been included in the evaluation fitting.

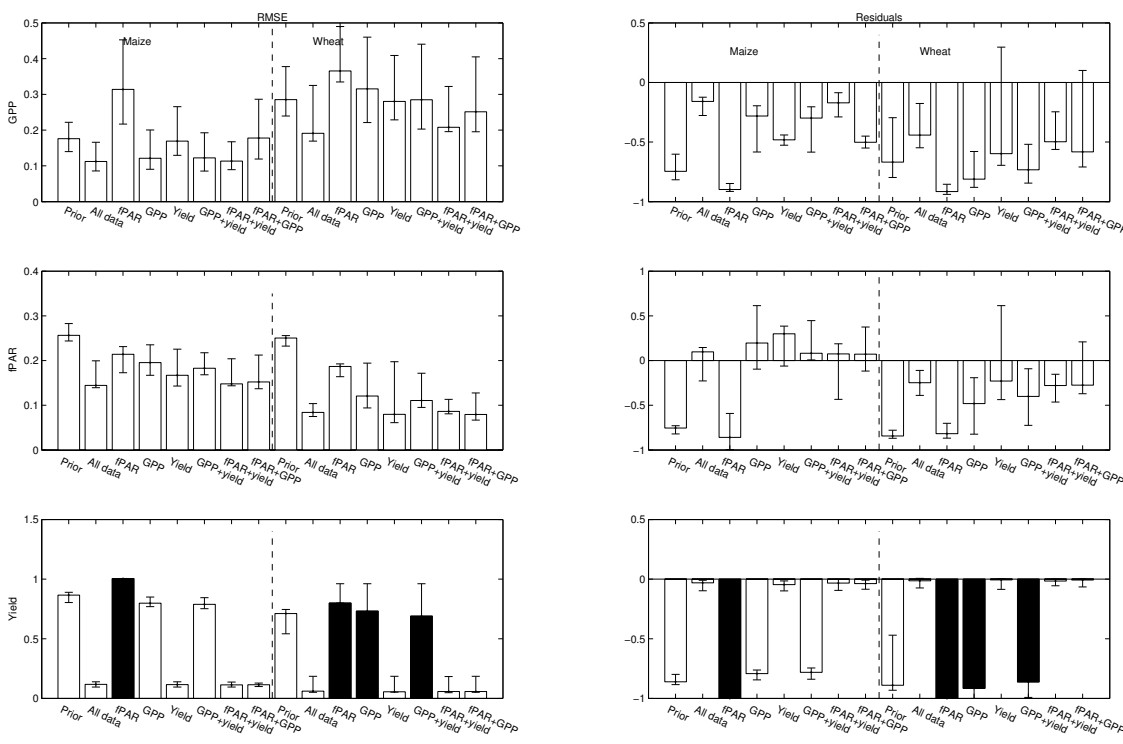

**Figure 5.** Model RMSE and bias for all data hold out experiments averaged over all wheat and maize sites respectively. Error bars represent variation across sites. All values have been normalised to the mean value of that variable at each site.Black bars indicate models that do not reach flowering.

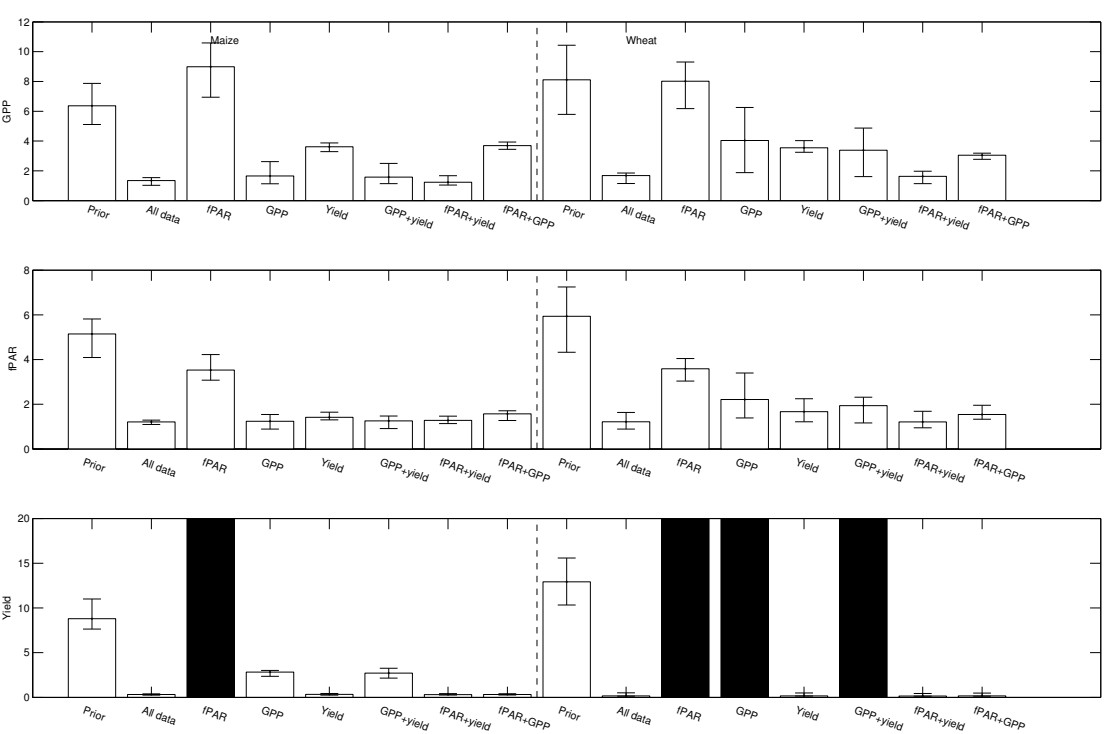

**Figure 6.** Model uncertainty, expressed as the difference between the upper and lower 95th confidence intervals for all model setups averaged across al wheat and maize sites. Error bars represent variation between sites. All values have been normalised. Black bars indicate models that do not reach flowering.

**Table 2.** Model parameters, upper and lower bounds and initial values used in the model fitting procedure

| Symbol | Units | Description | Lower bound | Upper bound | Initial value | Fixed |
|--------|-------|-------------|-------------|-------------|---------------|-------|
| $germ_{lim}$ | °C | Number of degree days required for germination | 100.0 | 400.0 | 150.0 | no |
| $Tb_{germ}$ | °C | Base temperature for germination | - | - | 0.0 | yes |
| $\rho$ | - | Optimal carbon to nitrogen ratio in vegetative tissue | - | - | 25.0 | yes |
| $N_0$ | g | Initial N content of the soil | 10.0 | 100.0 | 15.0 | no |
| $\theta$ | g N g$^{-1}$N g$^{-1}$ C day$^{-1}$ | Root nitrogen extraction factor | 0.0005 | 0.01 | 0.0005 | no |
| $V_{cmax25}$ | $\mu$mol m$^{-2}$ s$^{-1}$ | Photosynthetic carboxylation capacity at 25°C | 50.0 | 400.0 | 80.0 | no |
| $fJ$ | - | Ratio of electron transport to carboxylation capacity at 25°C | 2.0 | 10.0 | 2.1 | np |
| $\lambda_0$ | - | Ratio of atmospheric and leaf $CO_2$ concentration | 50.0 | 400.0 | 80.0 | no |
| $LMA$ | g m$^{-2}$ | Leaf mass per area | 60.0 | 400.0 | 100.0 | no |
| $r_{tot}$ | g g$^{-1}$ | Average plant respiration rate | 0.001 | 0.3 | 0.1 | no |
| $m_{trans}$ | g day$^{-1}$ | Mass translocation rate from vegetative to reproductive tissue | 0.1 | 20.0 | 2.0 | no |
| $C_{frac}$ | - | Carbon fraction of reproductive tissue | 0.2 | 1.0 | 0.7 | no |
| $p$ | days | Time period for averaging environmental conditions for flowering trigger | 1.0 | 30.0 | 10.0 | no |

**Table 3.** Model RMSE, bias and uncertainty for the one site and all site parametrisation as well as the model evaluation run

| | RMSE GPP | RMSE fAPAR | RMSE yield | Bias GPP | Bias fAPAR | Bias yield | Uncertainty GPP | Uncertainty fAPAR | Uncertainty yield |
|---|---|---|---|---|---|---|---|---|---|
| **Maize** | | | | | | | | | |
| **Prior** | 0.18 | 0.27 | 0.83 | -0.77 | -0.79 | -0.82 | 7.07 | 5.15 | 9.87 |
| **One site** | 0.11 | 0.14 | 0.12 | -0.16 | 0.10 | -0.03 | 1.34 | 1.21 | 0.33 |
| **All sites** | 0.08 | 0.16 | 0.10 | -0.12 | -0.04 | -0.00 | 0.45 | 1.02 | 0.12 |
| **Evaluation** | 0.09 | 0.15 | 0.11 | -0.10 | -0.09 | -0.00 | 0.47 | 1.08 | 0.15 |
| **Wheat** | | | | | | | | | |
| **Prior** | 0.27 | 0.25 | 0.67 | -0.64 | -0.83 | -0.83 | 7.92 | 5.46 | 12.27 |
| **One site** | 0.19 | 0.08 | 0.06 | -0.44 | -0.25 | -0.01 | 1.68 | 1.21 | 0.16 |
| **All sites** | 0.17 | 0.08 | 0.07 | -0.21 | 0.02 | 0.02 | 0.51 | 0.45 | 0.06 |
| **Evaluation** | 0.17 | 0.09 | 0.07 | -0.05 | -0.26 | 0.02 | 0.75 | 0.89 | 0.08 |

**Table 4.** RMSE, bias and uncertainty values the data knock out experiments for wheat and maize.

| Data fitted to | RMSE GPP | RMSE fAPAR | RMSE yield | Bias GPP | Bias fAPAR | Bias yield | Uncertainty GPP | Uncertainty fAPAR | Uncertainty yield |
|---|---|---|---|---|---|---|---|---|---|
| **Maize** | | | | | | | | | |
| **Prior** | 0.18 | 0.26 | 0.85 | -0.75 | -0.79 | -0.85 | 6.91 | 5.19 | 9.25 |
| **All data** | 0.11 | 0.14 | 0.12 | -0.16 | 0.10 | -0.03 | 1.34 | 1.21 | 0.33 |
| **fAPAR** | 0.31 | 0.21 | 1.00 | -0.90 | -0.86 | -1.00 | 8.99 | 3.53 | - |
| **GPP** | 0.12 | 0.20 | 0.80 | -0.28 | 0.20 | -0.79 | 1.66 | 1.24 | 2.83 |
| **yield** | 0.17 | 0.17 | 0.12 | -0.48 | 0.30 | -0.05 | 3.61 | 1.42 | 0.33 |
| **GPP+yield** | 0.12 | 0.18 | 0.79 | -0.30 | 0.08 | -0.78 | 1.58 | 1.26 | 2.72 |
| **fAPAR+yield** | 0.11 | 0.15 | 0.11 | -0.17 | 0.07 | -0.03 | 1.23 | 1.28 | 0.31 |
| **fAPAR+GPP** | 0.18 | 0.15 | 0.11 | -0.50 | 0.07 | -0.04 | 3.69 | 1.57 | 0.32 |
| **wheat** | | | | | | | | | |
| **Prior** | 0.28 | 0.25 | 0.70 | -0.66 | -0.84 | -0.88 | 8.49 | 5.90 | 12.98 |
| **All data** | 0.19 | 0.08 | 0.06 | -0.44 | -0.25 | -0.01 | 1.68 | 1.21 | 0.16 |
| **fAPAR** | 0.37 | 0.19 | 0.80 | -0.92 | -0.82 | -1.00 | 8.02 | 3.59 | - |
| **GPP** | 0.32 | 0.12 | 0.73 | -0.81 | -0.48 | -0.92 | 4.04 | 2.21 | - |
| **yield** | 0.28 | 0.08 | 0.06 | -0.60 | -0.23 | -0.01 | 3.55 | 1.67 | 0.16 |
| **GPP+yield** | 0.28 | 0.11 | 0.69 | -0.73 | -0.40 | -0.86 | 3.38 | 1.94 | - |
| **fAPAR+yield** | 0.21 | 0.09 | 0.06 | -0.50 | -0.28 | -0.02 | 1.63 | 1.21 | 0.16 |
| **fAPAR+GPP** | 0.25 | 0.08 | 0.06 | -0.58 | -0.27 | -0.01 | 3.05 | 1.55 | 0.16 |

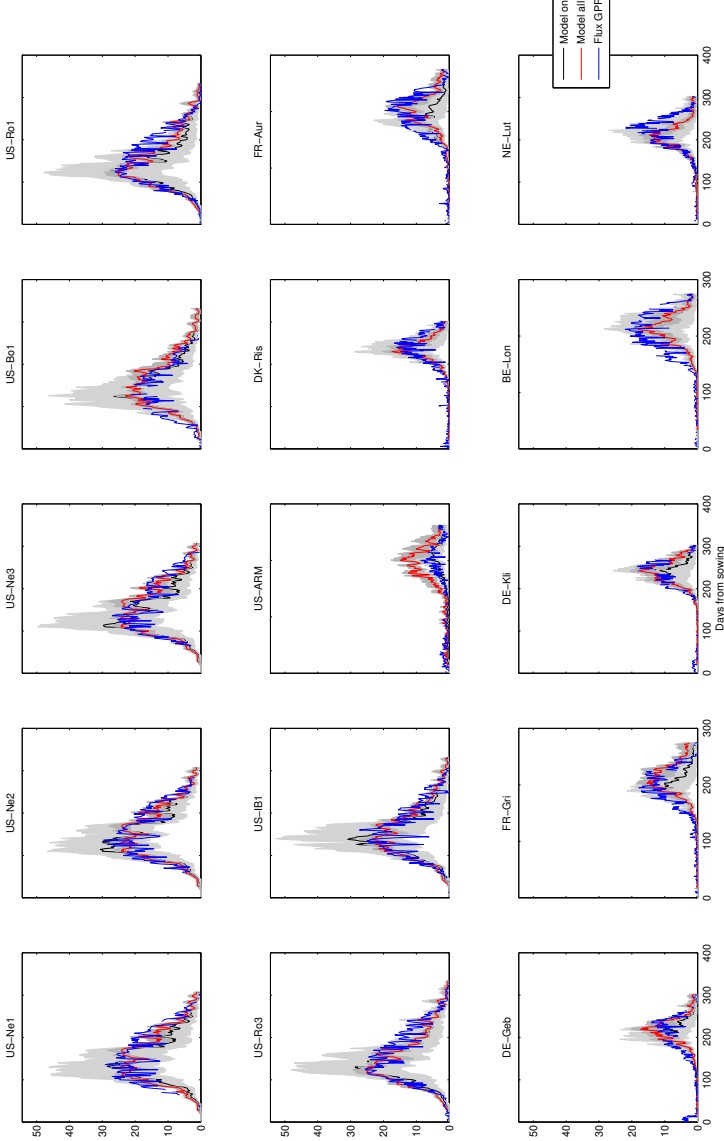

**Figure A1.** Gross primary production predictions for one year for all sites fitted using all available data at each individual site and at all sites together. Gray shaded areas represent 95% confidence intervals drawn from the posterior distribution.

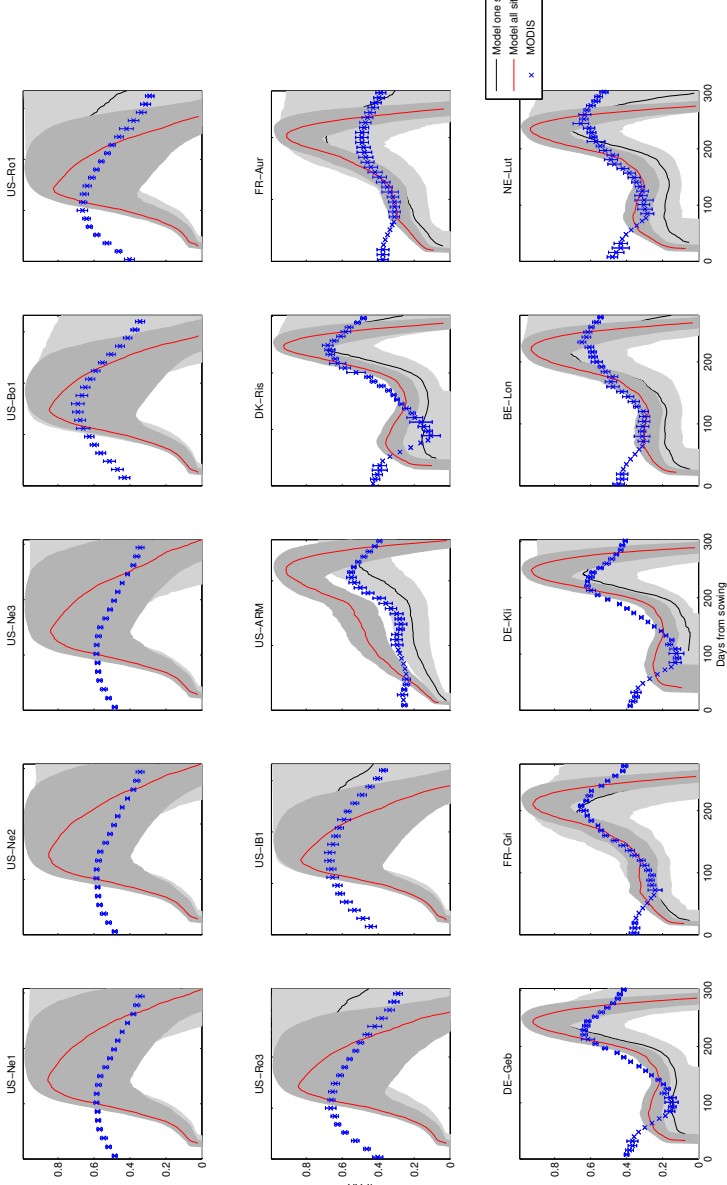

**Figure A2.** fAPAR predictions for one year for all sites fitted using all available data at each individual site and at all sites together. Gray shaded areas represent 95% confidence intervals drawn from the posterior distribution.

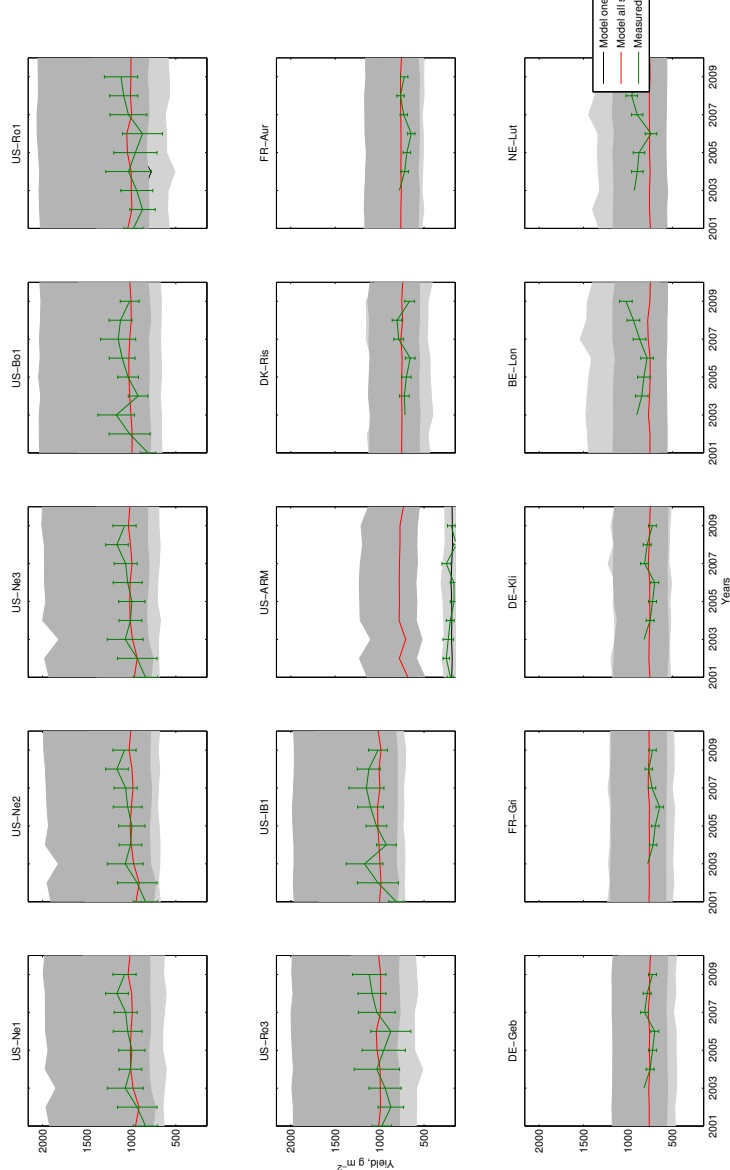

**Figure A3.** yield predictions for all years for all sites fitted using all available data at each individual site and at all sites together.Gray shaded areas represent 95% confidence intervals drawn from the posterior distribution.

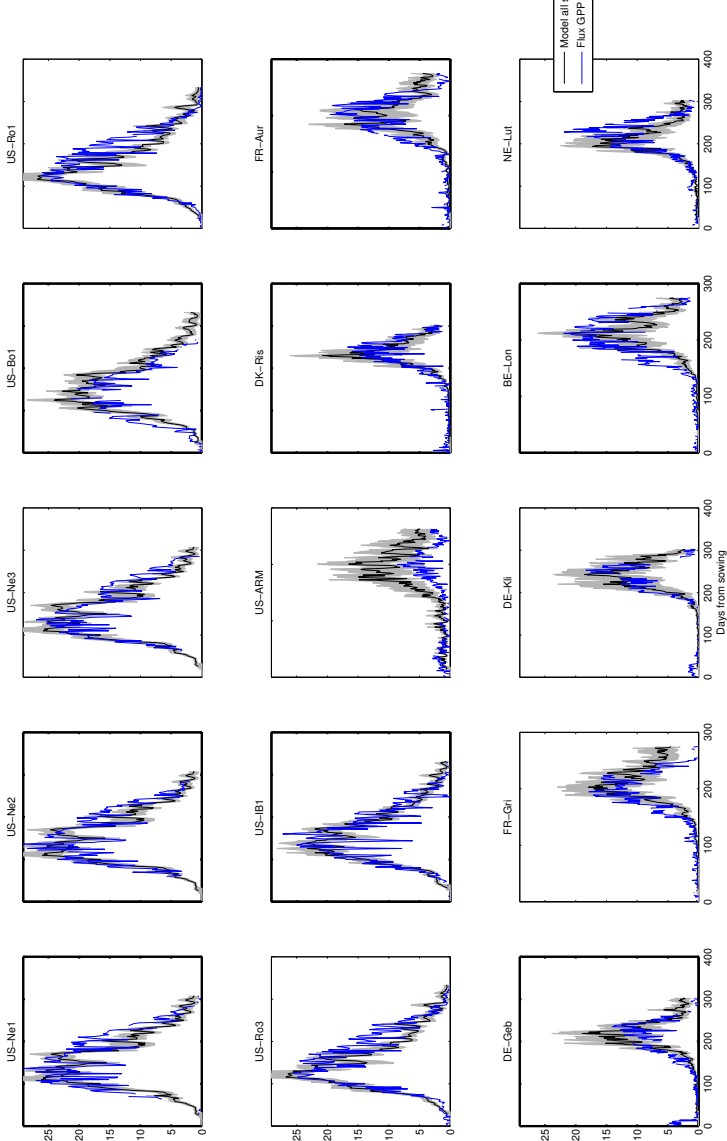

**Figure A4.** Gross primary production predictions for one year for all sites fitted using all available data at a subset of sites for model evaluation. Sites with black boxes have been used in the model fitting. Gray shaded areas represent 95% confidence intervals drawn from the posterior distribution.

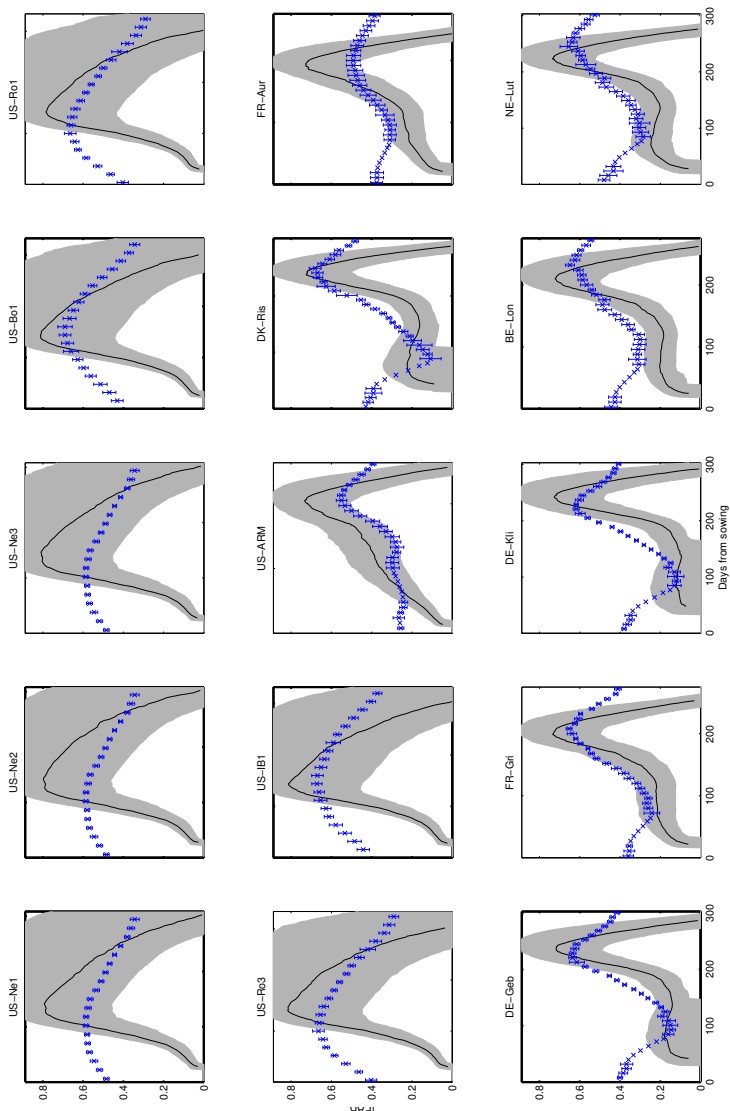

**Figure A5.** fAPAR predictions for one year for all sites fitted using all available data at a subset of sites for model evaluation. Sites with black boxes have been used in the model fitting. Gray shaded areas represent 95% confidence intervals drawn from the posterior distribution.

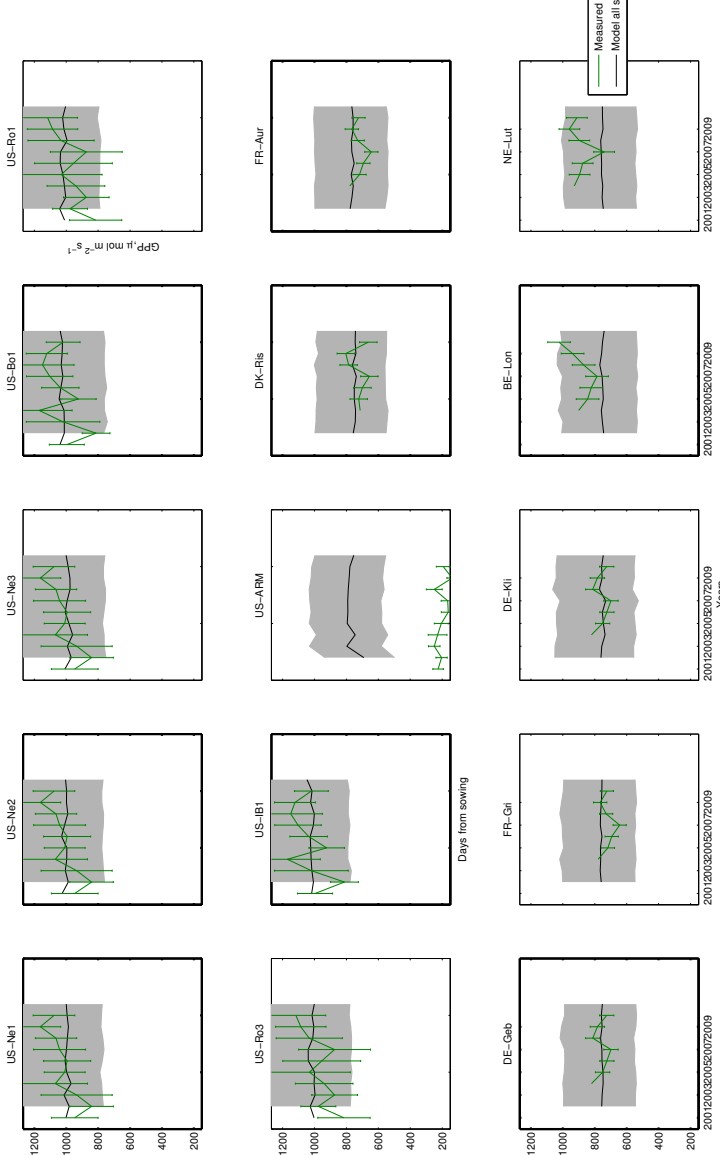

**Figure A6.** yield predictions for all years for all sites fitted using all available data at a subset of sites for model evaluation. Sites with black boxes have been used in the model fitting. Gray shaded areas represent 95% confidence intervals drawn from the posterior distribution.

**Table A1.** Photosynthesis model constants according to dePury and Farquhar (1997) for C3 photosynthesis and adapted from Haxeltine et al. (1996) for C4 photosynthesis.

| Symbol | Units | Description | C3 value | C4 value |
|--------|-------|-------------|----------|----------|
| $\Gamma_*$ | Pa | $CO_2$ compensation point | 3.69 | 0 |
| $K'$ | Pa | Effective Michaelis-Menten constant of Rubisco | 73.8 | 94.61 |
| $\Theta$ | - | Curvature of leaf response of electron transport to irradiance | 0.7 | 0.7 |
| $f$ | - | Spectral correction factor | 0.15 | 0.15 |