# Peer review of "The impacts of data constraints on the predictive performance of a general process-based crop model (PeakN-crop v1.0)"

_Geoscientific Model Development, 2016_

## Short Comment (SC1) · 15 Nov 2016

General comments

This paper proposes and tests a new approach to estimation of crop yields applicable to sites with available remote sensing data, tower eddy covariance data and regional yield. The overall framework is the quest for a prediction method that is applicable at many or ultimately most sites around the world, a very ambitious goal. The authors correctly argue that the availability of data is the major constraint, and here they innovatively combine data sets which are available at many worldwide sites.

A second challenge is to develop a model that can use the available data as inputs

for prediction. The authors argue that process based models have major advantages compared to statistical models for extrapolating to weather conditions that might be experienced under climate change, but that current process based models are not sufficiently general for worldwide use. They therefore propose a new, simple process based model which is supposed to have sufficient generality. The model is based on assuming that plants optimize the root:shoot ratio and also the time of flowering. Finally, the authors test their approach across 15 sites and two crops, wheat and maize.

I think that this is an important paper, because of the issues raised and the innovativeness of the proposed solutions. There are many possible criticisms, and I detail a certain number below. That is, the authors have not found a satisfactory answer to the overall problem of predicting yields at arbitrary global locations. That is hardly surprising given the magnitude of the objective, and should not obscure the real contributions of the study.

Specific comments

A first criticism is that there seems to be some ambiguity about the exact objective of the predictions. In the introduction the authors speak of application to a "generic farm location", whereas evaluation is based on comparison with regional yields. Prediction for a farm, with uniform management, is quite different than prediction for a region. The paper seems more oriented toward regional prediction, since county yields are one of the data sets used as input and are also used for evaluation. On the other hand, the landcover input data was aggregated to 3km by 3 km pixels, which is generally intermediate between farm and county scale. In any case, it is essential to clarify the spatial scale of interest.

Much of the evaluation is based on comparing the model using data constrained parameters to the model with prior parameters. This is not a very interesting comparison. The prior parameters were chosen quite arbitrarily by the authors to represent essentially a total lack of information about the parameter values. The fact that adding some

information improves the situation is hardly surprising. A much more relevant comparison would be between the data constrained model and long term average county yields. Does the model do better than simply assuming that the future is like the average of the past? This is analogous to comparing climate forecasts with climatology.

Another aspect of the evaluation is that uncertainty intervals are given for each prediction. This is extremely informative and pertinent, and is a very valuable addition to the comparison between the mean prediction and observed values. However, the uncertainty results need to be discussed more thoroughly. For example, it seems that the 95% intervals for yield cover all historic yields at most sites (Fig. 3a). Surely this uncertainty is so large as to render the results useless. More discussion is required here.

The uncertainty calculations are based on propagating uncertainty in the parameter values through the model. It is not clear if residual error is included when calculating uncertainty intervals or not. It should be. Also, there are other sources of uncertainty than the parameters which might be quite important, in particular uncertainty about management practices. This should at least be discussed.

It seems that the likelihood used here for the Bayesian estimation assumes that all data are independent. This is of course almost certainly false for time series data. Taking non independence into account by dividing by the number of measurements is only a very crude approximation.

The model that is proposed is an original model, based on the assumption that plants optimize partitioning between roots and aboveground biomass, as well as time of flowering. The major advantage of such a model is that it allows the same model, with the same parameters, to be used for different cultivars of the same species, if one accepts that the cultivars chosen for a particular location are optimized for the environment there. More detail about the model would be helpful. How exactly is the date of flowering calculated? According to the text, the switch from vegetative to reproduc-
tive growth occurs when increased vegetative fractions would not result in an overall increase in growth rate. Is this calculated day by day or is there some averaging over environmental conditions to ensure that the plant doesn't respond to conditions on one specific day?

What exactly are the management inputs required for the model? The authors mention sowing and harvest date, but aren't sowing density and fertilizer inputs also required? The required management information should be made clear, as well as the sources of this information.

There is also no information on soils. Apparently this information is not needed here thanks to the assumptions that there is no water limitation, and that initial soil N is negligible compared to fertilizer N. In general, however, it will be necessary to have soil information.

The authors suggest that the model could be tested by comparing different model structures. Perhaps more useful would be to test the model proposed here with much more detailed input data, in order to reduce the data as a source of error and thereby isolate the amount of error due to the model.

Technical corrections

P7 L23-24. "given the model" needs to be omitted
* * *

---

## Referee Comment (RC2) · Anonymous Referee #2 · 3 Jan 2017

The manuscript touches an interesting topic; however, the coverage of it is unsatisfying yet. Introduction The introduction could be much more focused. The extensive linkage to 'food security' is not necessary and the meaning of the own contribution overstated. The use of categories is not convincing. How can statistical models be considered non-mathematical and process based models mathematical? (line 16-17). The influence of the different data sources on parametrization is not considered. For example, the inclusion of farm yield data would necessarily imply that management effects influence the parametrization. This is similar to the parametrization of statistical models and should have been addressed in a different way on page 2 lines 30-35. The introduction ends with three valid research questions, however, the concrete model that will

be used to address these questions remains open. Material and Methods Instead of using an established crop model in their exercise the authors claim the development of a 'general process-based crop model' which is then used in an initial set-up and in different forms of a data constrained mode. The claim for a new model (page 5 line 15) adds surprising additional dimensions to the paper. The description of the model is totally unsatisfying. The presentation of the equations does not follow a systematic scheme. Essential information is missing. How was the soil variability parametrized? The original parameter values are not given and any validation results are missing. As one consequence prior expectations about the parameter values cannot be given. The assigned uncertainties for the given data sources are difficult to follow. A systematic reasoning for the chosen uncertainty values is missing. Results The presentation of the results continues the deficits of the M&M section. It does not fulfill the existing standards. It is like an experimental paper that gives information about explained and unexplained variability, but no information about the means and the quantitative extend of effects. What was the quantitative propagation of the initial parameter setting? This leads to my main criticism of the paper: the results given are not reproducible. In the current form the paper is the technical description of an achieved status, it has not yet reached the maturity for publication in a scientific journal.

---

## Author Comment (AC1) · 30 Jan 2017

Response to Reviewer 1

First of all we would like to thank the reviewer for the very thorough analysis and constructive comments. We believe that addressing these comments will add value to our work. We would also like to thank him for signing the review, which is an important part of the open review process.

Below are our responses to the specific comments made, including how we intend to address them with specific changes in the manuscript.

Comment from reviewer 1: "A first criticism is that there seems to be some ambiguity

about the exact objective of the predictions. In the introduction the authors speak of application to a "generic farm location", whereas evaluation is based on comparison with regional yields. Prediction for a farm, with uniform management, is quite different than prediction for a region. The paper seems more oriented toward regional prediction, since county yields are one of the data sets used as input and are also used for evaluation. On the other hand, the landcover input data was aggregated to 3km by 3 km pixels, which is generally intermediate between farm and county scale. In any case, it is essential to clarify the spatial scale of interest."

Response from Authors 1: The model is intended, in an ideal situation, to be used at the local scale. However, given the lack of availability of the data, particularly the yield information, this is difficult. The model as applied here is indeed at an intermediate scale, as the eddy flux data is always field scale, while the crop yield is regional. We agree that this discrepancy between the different spatial scales in the data has not been discussed sufficiency and we have now added this o the discussion.

Proposed changes to Manuscript 1: We will add to the end of the introduction a clarifying statement about the scale of application, such as

"While our study is part of a boarder scientific objective to enable more accurate field scale predictions, the lack of availability of field scale datasets to train and validate our model means that the scale of model evaluation for our study here is a mix of field (flux tower) and regional scales (county and country level for yield estimates and 3 by 3 km scale for photosynthetic activity)."

And we will add a dedicated paragraph to the discussion about this issue

"One additional complication is the different spatial scales of the three datasets - while the eddy covariance data is at the scale of the flux tower footprint, which can be seen as equivalent to the individual field scale, the fAPAR and yield data correspond to larger scales (county and country level for the yield data and a 3 by 3 km scale for the fAPAR data). The assumption behind our analysis is that the conditions at field

scale are representative of the regional scale, so that there would be no discrepancy between model predictions at these different scales. This is obviously a source of error, especially at the wheat sites in Europe, which will be located over a much more heterogeneous landscape. Further sources of data at the field scale would be required to identify the model error caused by the discrepancy in spatial scales."

Comment from Reviewer 2: "Much of the evaluation is based on comparing the model using data constrained parameters to the model with prior parameters. This is not a very interesting comparison. The prior parameters were chosen quite arbitrarily by the authors to represent essentially a total lack of information about the parameter values. The fact that adding some information improves the situation is hardly surprising."

Response from Authors 2: This is indeed a not very surprising result. However, we have chosen the model with prior parameters as a benchmark as this offers a worst case scenario of parameter values and can be used to show relative improvement in model error and uncertainty when using the different datasets.

Proposed changes to Manuscript 2: We will include a statement in the introduction to reflect on the fact that we fully expect data constraining the model to improve predictive accuracy but that we're more interested in how much the predictions are improved. We propose to insert just after we state the paper aims:

"We expect the qualitative answer to the first question to be that utilising empirical data does enable the model to make better predictions because that's a typical outcome of our parameter estimation approach. However we are more interested in the quantitative answer; i.e. how much? For example, the generation of a model that could make extremely precise and accurate predictions would suggest that data-constraining general models with the datasets we identify could provide an extremely useful tools for agricultural predictions and forecasts. Alternatively, the generation of a model that makes very imprecise predictions would suggest that more data collection and model improvement is needed for the model to have practical applications."

[Figure]

Comment from Reviewer 3: "A much more relevant comparison would be between the data constrained model and long term average county yields. Does the model do better than simply assuming that the future is like the aver-age of the past? This is analogous to comparing climate forecasts with climatology.

Response from Authors 3: This is a very good idea, however, we consider that such a comparison with long terms yields is premature. We have not performed this analysis for two reasons: inadequate data and the mix of spatial scales. As mentioned our ultimate objective is to enable better field scale predictions, however the only field scale data we have, the flux datasets, typically only cover one year and different sites cover different years. This makes comparing model predictions to a data average for one site almost meaningless. Our other datasets do have longer, more complete time series. The fAPAR time series reflects vegetation dynamics over a wider spatial area of vegetation than an individual field, including non-agricultural vegetation and so itself is only a coarse indicator of field level dynamics - we are therefore not interested in how well the model captures the complete seasonal dynamics of that dataset. It would be interesting to understand whether a model such as that developed here can out-perform historical data at predicting large scale yield variations. This should be dealt with in a dedicated future piece of work because a number of other analyses should be conducted alongside such a comparison to make it insightful. For example we currently do not account for variations in environmental drivers and agricultural practices at smaller spatial scales across a region that would also help to explain yield variations observed at the larger spatial scales. These could be accounted for by simulating our model for every relevant field within the region and aggregating the predictions. A dedicated comparison of the predictions of our model with those of statistical averages over historical data is a natural avenue for future research.

Proposed changes to Manuscript 3: We have thought hard about how to implement this request but, as explained above, evaluation for each dataset raises further issues. We have therefore decided to defer this comparison to another study, ideally one in which

our model can be trained and evaluated against a richer dataset of location specific data. We propose to include a brief statement in the discussion (under future data needs) stating

"If this model, or any other similar process-based data constrained crop model, is to be used for scientific purposes to understand the response of crops to climate across the globe, the ideal data would be a global data set, such as space-based vegetation observations, combined with high quality field level data that would ideally include growth timeseries, final grain yield and information about agricultural practices. However, if the model is to be used for agricultural purposes, to aid decision making at the local level, high quality field level data would be sufficient. A valuable evaluation in such studies, not conducted here for brevity and due to a lack of location-specific data, would be to compare the predictive accuracy of the model against the predictive accuracy of a statistical average over the data. Such an analysis would reveal whether and how much benefit is gained by using a data constrained model for predictions"

Comment from Reviewer 4: "Another aspect of the evaluation is that uncertainty intervals are given for each prediction. This is extremely informative and pertinent, and is a very valuable addition to the comparison between the mean prediction and observed values. However, the uncertainty results need to be discussed more thoroughly. For example, it seems that the 95% intervals for yield cover all historic yields at most sites (Fig. 3a). Surely this uncertainty is so large as to render the results useless. More discussion is required here."

Response from Authors 4: We agree that the uncertainty in yield predictions is very large, making it largely unsuitable for predicting interannual variation in crop yields. However, as the reviewer also points out, the model parametrisation method presented in our paper has the advantage that is does provide an uncertainty estimate. Hence, any future improvements in model performance, especially through added sources of data are easily quantifiable.

Proposed changes to Manuscript 4: We will expand an existing paragraph in the discussion to read

"Model uncertainty is difficult to compare with previous crop modelling studies, as models with fixed parameter values do not often provide uncertainty estimates. In fact, providing uncertainty values for all model variables and parameters is one of the advantages of a data constrained model. In the current model, uncertainty is highest at the start of the season for all variables but decreases rapidly and final yield uncertainty is much lower. This is due to thresholds: abrupt changes from one growing stage to another when small differences in parameters can lead to large differences in resulting variables. It is, however, important to note that the uncertainty in our yield predictions remains high and the model in its current form is unlikely to provide accurate predictions for practical applications without the addition of new data (Section 7.4). We have however shown that the use of three different data types does reduce prediction uncertainty - pointing to an avenue for future model improvement."

Comment from Reviewer 5: "The uncertainty calculations are based on propagating uncertainty in the parameter values through the model. It is not clear if residual error is included when calculating uncertainty intervals or not. It should be."

Response from Authors 5: The model uncertainty is calculated by sampling parameters from the posterior distributions and then computing model predictions with the sampled parameters, which results in a distribution of model predictions from which we can calculate a predicted mean and confidence intervals. We realise that this was insufficiently well explained clearly in the paper.

Proposed changes to Manuscript 5: We will include a fuller explanation of how we propagate uncertainty in the parameter values of the model:

"To calculate uncertainty for the model predictions we sample parameter values from their respective posterior distribution and compute predictions with each parameter combination, which results in a corresponding distribution of model predictions. We

report this prediction distribution uncertainty using 95th percent confidence intervals. This predicted distribution does not include the prescribed or inferred uncertainty about observations, $\sigma_{}(x,D)$, our predicted distributions correspond to the state being predicted and not the observations of that state."

Comment from Reviewer 6: "Also, there are other sources of uncertainty than the parameters which might be quite important, in particular uncertainty about management practices. This should at least be discussed."

Response from Authors 6: We agree and will make a change to the manuscript to reflect this.

Proposed changes to Manuscript 6: We will include a discussion of model uncertainty related to management practices, in particular sowing and harvest dates as well as fertilizer input, such as

"In addition to the three datasets used for parametrisation, the model also requires input data in the form of sowing and harvest dates and fertiliser inputs. Additional uncertainty is associated with these datasets which is not available nor accounted for in our analyses. For example, the crop calendar (Sacks et al., 2010) and Nitrogen Fertilizer Application (Potter et al., 2010) datasets are global data collections that will imperfectly represent the value for any given location. Alternatives to these global datasets would be to use location-specific data, or to infer the values. Location specific data has the advantage of more accurately reflecting the situation at a given site and would therefore be useful when the model is applied at the field scale, but such data is unlikely to be available for all sites. Successful inference of the values would depend on if there is enough information in the datasets used to infer the model parameters. If there is inadequate data then there would be excessive degrees of freedom for inference, leading to the wrong parameter values begin inferred and the model performing poorly in novel situations. Therefore, the decision whether to obtain more data or infer unknown quantities in future applications of our model and inference framework depends on the

data availability and the intended scales of application."

Comment from Reviewer 7: "It seems that the likelihood used here for the Bayesian estimation assumes that all data are independent. This is of course almost certainly false for time series data. Taking non independence into account by dividing by the number of measurements is only a very crude approximation."

Response from Authors 7: The division by the number of measurements is not meant to account for non independence, rather it accounts for the different number of data points in each time series so that each dataset is given equal importance for fitting purposes. We agree that the timeseries data is most likely not independent but the independence assumption is often used when fitting models to eddy covariance flux data in order to simplify the formulation of the likelihood function.

Proposed changes to Manuscript 7: We will include the point made by the reviewer after we describe our Likelihood function.

"Note that with this definition of the likelihood we are treating every data point as independent, that is the likelihood of a value at time ,t, is treated independently from the likelihoods at preceding times. This is only an approximation but is commonly used in parameter estimation studies because the additional mathematical and computational complexity of accounting for non-independent data."

Comment from Reviewer 8: "More detail about the model would be helpful. How exactly is the date of flowering calculated? According to the text, the switch from vegetative to reproductive growth occurs when increased vegetative fractions would not result in an overall increase in growth rate. Is this calculated day by day or is there some averaging over environmental conditions to ensure that the plant doesn't respond to conditions on one specific day?"

Response from Authors 8: The start of the reproductive growth is calculated using an average of environmental conditions for the peak carbon criteria. Due to the continuous

decrease in soil N driven only by plant uptake and the relative simplicity of the model which does not take temperature and moisture into account for nutrient uptake, this averaging is not necessary for the peak N criteria. We will clarify this in the text

Proposed changes to Manuscript 8: We will include a statement in the methods about this:

"The peak nitrogen condition is achieved when an increase in root mass does not result in an increase in nitrogen uptake. This condition is achieved in nitrogen limited environments where the nitrogen available in the soil is depleted through the period of vegetative growth. This assumption can be considered valid in agricultural systems where the major nitrogen input into the system during the growing period comes solely from agricultural fertilisers. Soil nitrogen decays monotonically through the season in our model due to the simplicity with which we model nitrogen uptake and so detecting the peak nitrogen condition is straightforward. Similarly, the peak carbon flowering condition is triggered when the addition of aboveground biomass would not lead to an increase in net carbon gain, due to self-shading in the canopy. To calculate the peak carbon trigger we use the environmental variables averaged over p days, to avoid flowering being triggered by short-term environmental fluctuations. We infer p alongside the other parameters in our model."

Comment from Reviewer 9: "What exactly are the management inputs required for the model? The authors mention sowing and harvest date, but aren't sowing density and fertilizer inputs also required? The required management information should be made clear, as well as the sources of this information."

Response from Authors 9: This is correct, in addition to sowing and harvest dates, fertilizer input and planting density are also required, as well as irrigation regimes for future versions of the model which would take into account water limitation. We will include more information on this, including sources of the values used on the study.

Proposed changes to Manuscript 9: We propose to include these details in the methods

"Fertilizer input data were obtained from the published site descriptions (see Table 1 for references) or from the Nitrogen Fertilizer Application database (Potter et al. 2010). The model implemented in this study does not require any additional information on irrigation or soil properties."

Comment from Reviewer 10: "There is also no information on soils. Apparently this information is not needed here thanks to the assumptions that there is no water limitation, and that initial soil N is negligible compared to fertilizer N. In general, however, it will be necessary to have soil information."

Response from Authors 10: Additional soil information would be essential for versions of the model that include water limitation and very important if the N uptake model was made more complex, for example if we included information about rooting depth or different forms of available nitrogen.

Proposed changes to Manuscript 10: We will include a paragraph on this point in the discussion

"The model in the version presented in this paper does not include any water limitation to growth due mainly to a lack of data constraint on any water related parameters, as we found that latent heat data from EC towers is not sufficient. Below-ground measurements of not only root growth but also soil water properties would again provide some of the necessary information. Such belowground data, especially if supplemented by nutrient concentrations can also help constrain a more complex version of the nitrogen uptake scheme, which could be improved to include more explicit soil-plant interactions and additional processes such as biological nitrogen fixation for legumes."

Comment from Reviewer 11: "The authors suggest that the model could be tested by comparing different model structures. Perhaps more useful would be to test the model proposed here with much more detailed input data, in order to reduce the data as a source of error and thereby isolate the amount of error due to the model."

Response from Authors 11: Comparing different model structures and including better quality data are two different, but valid ways of identifying sources of error in our analysis and the two combined would give the most accurate analysis.

Proposed changes to Manuscript 11: We believe this point is already covered by one of the final statements in our Discussion

"If this model, or any other similar process-based data constrained crop model, is to be used for scientific purposes to understand the response of crops to climate across the globe, the ideal data would be a global data set, such as space-based vegetation observations, combined with high quality field level data that would ideally include growth timeseries, final grain yield and information about agricultural practices. However, if the model is to be used for agricultural purposes, to aid decision making at the local level, high quality field level data would be sufficient."

Comment from Reviewer 12: "P7 L23-24. "given the model" needs to be omitted"

Proposed changes to Manuscript 12: This will be corrected.

―――――――――――――――――

---

## Author Comment (AC2) · 30 Jan 2017

Response to Reviewer 2

We thank the reviewer for their review. We believe that addressing these comments will add value to our work. We did find some of the comments unclear but have done our best to reply to them all.

Comment from Reviewer 1: "The extensive linkage to 'food security' is not necessary and the meaning of the own contribution overstated."

Response from Authors 1: Since GMD is not a subject specific journal we find it is helpful to include the link to the bigger picture and clarify why improving agricultural

models is important. The link to food security was the main reason why we undertook the research described in the manuscript.

Proposed changes to Manuscript 1: We have not removed our original framing of the study. However, in response to comments from the other reviewer, we proposed to include more discussion on the lack of precision of our current model at predicting crop yields and the need for more research with more location-specific data, such as

"Model uncertainty is difficult to compare with previous crop modelling studies, as models with fixed parameter values do not often provide uncertainty estimates. In fact, providing uncertainty values for all model variables and parameters is one of the advantages of a data constrained model. In the current model, uncertainty is highest at the start of the season for all variables but decreases rapidly and final yield uncertainty is much lower. This is due to thresholds: abrupt changes from one growing stage to another when small differences in parameters can lead to large differences in resulting variables. It is, however, important to note that the uncertainty in our yield predictions remains high and the model in its current form is unlikely to provide accurate predictions for practical applications without the addition of new data (Section 7.4). We have however shown that the use of three different data types does reduce prediction uncertainty - pointing to an avenue for future model improvement."

and

"If this model, or any other similar process-based data constrained crop model, is to be used for scientific purposes to understand the response of crops to climate across the globe, the ideal data would be a global data set, such as space-based vegetation observations, combined with high quality field level data that would ideally include growth timeseries, final grain yield and information about agricultural practices. However, if the model is to be used for agricultural purposes, to aid decision making at the local level, high quality field level data would be sufficient. An valuable evaluation in such studies, not conducted here for brevity and due to a lack of location-specific data, would be to

compare the predictive accuracy of the model against the predictive accuracy of a statistical average over the data. Such an analysis would reveal whether and how much benefit is gained by using a data constrained model for predictions"

Comment from Reviewer 2: "The use of categories is not convincing. How can statistical models be considered non-mathematical and process based models mathematical? (line 16-17)."

Response from Authors 2: The phrasing on lines 16-17 is indeed wrong. The process-based and statistical model separation is one that is commonly used not only for crop models but also in the field of earth system models and one that we find useful in explaining how process knowledge and data are used to obtain agricultural predictions.

Proposed changes to Manuscript 2: We will edit the confusing sentence to read

"Predicting and understanding how crops respond to changes in their environment through the use of mathematical models is needed to help address such threats, enabling advanced warning of potential threats and predictions of what alterations to agricultural practices might help prevent or mitigate problems."

We then go on to explain in detail the difference between process-based and statistical models (both are mathematical!)

Comment from Reviewer 3: "The influence of the different data sources on parametrization is not considered. For example, the inclusion of farm yield data would necessarily imply that management effects in-fluence the parametrization. This is similar to the parametrization of statistical models and should have been addressed in a different way on page 2 lines 30-35.

Response from Authors 3: The influence of different data sources on model parametrization is the main topic of our paper. management and field level information is required in process based models but not included explicitly in statistical models, as we discuss in the paragraph mentioned by the reviewer. Unfortunately the meaning of

this comment is not entirely clear.

Proposed changes to Manuscript 3: we now include more details of where we got our data for fertilizer, sowing and harvest dates – an issue also raised by the other Reviewer.

"In addition to the three datasets used for parametrisation, the model also requires input data in the form of sowing and harvest dates and fertiliser inputs. Additional uncertainty is associated with these datasets which is not available nor accounted for in our analyses. For example, the crop calendar (Sacks et al., 2010) and Nitrogen Fertilizer Application (Potter et al., 2010) datasets are global data collections that will imperfectly represent the value for any given location. Alternatives to these global datasets would be to use location-specific data, or to infer the values. Location specific data has the advantage of more accurately reflecting the situation at a given site and would therefore be useful when the model is applied at the field scale, but such data is unlikely to be available for all sites. Successful inference of the values would depend on if there is enough information in the datasets used to infer the model parameters. If there is inadequate data then there would be excessive degrees of freedom for inference, leading to the wrong parameter values begin inferred and the model performing poorly in novel situations. Therefore, the decision whether to obtain more data or infer unknown quantities in future applications of our model and inference framework depends on the data availability and the intended scales of application."

Comment from Reviewer 4: "The introduction ends with three valid research questions, however, the concrete model that will be used to address these questions remains open."

Response from Authors 4: We should indicate in the introduction that we intend to introduce and use a new model.

Proposed changes to Manuscript 4: We will clarify in the introduction that we use a new model

"In this paper we present a newly developed general, non-crop specific process based model and use parameter inference to infer the most likely parameters for 15 locations for winter wheat and maize using a combination of space-based vegetation indices, eddy covariance flux data and reported agricultural yields."

Comment from Reviewer 5: "The claim for a new model (page 5 line 15) adds surprising additional dimensions to the paper."

Response from Authors 5: As we discussed in section 7.3, we chose to use a new model as it is more general and allows us to perform our analysis for multiple sites and species. We acknowledge that the use of this new model also has certain disadvantages and we mention this in the discussion.

Proposed changes to Manuscript 5: We will partially address this by mentioning that the model is new in the introduction. We also already cover the need to compare our model to others in the discussion

"Here we have chosen a given model structure and extensively tested the way in which constraining the parameters with different datasets in different configurations. The question that arises is to what extent the chosen model itself affects the present results. We have chosen a novel, physiology based model which includes plant optimality concepts, which on one hand has the advantage that it is more general than some of the older models and lacks artificially set thresholds between growth stages, but does have the disadvantage of being less thoroughly tested against field observations. An ideal companion paper to this study would be a comparison of different model structures with a constant data constraining framework, providing greater insights into which parts of the model lead to high errors or uncertainty. However, given the limitations of the current study, we acknowledge this limitation and report most error metrics as relative to prior model runs in an attempt to isolate errors created by the data and model fitting from those caused by the model itself."

Comment from Reviewer 6: "How was the soil variability parametrized?"

Response from Authors 6: As this is a very simple model at this stage the only soil information needed was nitrogen fertilizer application.

Proposed changes to Manuscript 6: At the suggestion of both reviewers we have added a discussion of any additional soil information needed for a more detailed model.

"The model in the version presented in this paper does not include any water limitation to growth due mainly to a lack of data constraint on any water related parameters, as we found that latent heat data from EC towers is not sufficient. Below-ground measurements of not only root growth but also soil water properties would again provide some of the necessary information. Such belowground data, especially if supplemented by nutrient concentrations can also help constrain a more complex version of the nitrogen uptake scheme, which could be improved to include more explicit soil-plant interactions and additional processes such as biological nitrogen fixation for legumes."

Comment from Reviewer 7: "The original parameter values are not given and any validation results are missing."

Response from Authors 7: As we explain in section 4 (Parameter estimation technique) we use a Bayesian fitting method which requires prior intervals for the parameter but not prior parameter values. As explained in section 5, the prior parameter values are randomly sampled from the prior parameter distribution in a manner similar to parameters being sampled from the posterior. The paper contains extensive model validation, in fact it contains little else. Figure 1 shows a comparison of prior and posterior model performance and figures in the appendix contain site level model-data comparison as the results of cross-site validation. Model validation is discussed extensively in both the results and discussion section.

Proposed changes to Manuscript 7: We will adjust the aim statement in the paper to make clear that we are inferring our parameters

"In this paper we present a newly developed general, non-crop specific process based

model and use parameter inference to infer the most likely parameters for 15 locations for winter wheat and maize using a combination of space-based vegetation indices, eddy covariance flux data and reported agricultural yields."

Comment from Reviewer 8: "The assigned uncertainties for the given data sources are difficult to follow. A systematic reasoning for the chosen uncertainty values is missing."

Response from Authors 8: A description of how we include data uncertainty in model fitting can be found in section 4. We acknowledge that this can be difficult to follow for those new to, or unfamiliar with, with Bayesian fitting methods and we will extend this description

Proposed changes to Manuscript 8: We propose to adjust the paragraph on data uncertainty to read.

"We adopt different techniques to estimate the standard deviation $\sigma\_(x,D)$ above, depending on the dataset D at each location. Generally, we assume that the variation in the model predictions about the data is solely due to uncertainty in the data. The GPP data do not have an estimate of uncertainty and so we infer the uncertainty associated with those data as the parameter $\sigma\_(x,D)$. In the case of MODIS fAPAR data we explicitly incorporate a measure of variation in the data within the geographical area used to compute the mean fAPAR as well as inferring a parameter representing additional unexplained variation. We include this parameter to account for known issue in space based remotely sensed data, such as background soil reflectance. The crop yield data already have estimates of observational uncertainty associated with them and so we use those data to define $\sigma\_(x,D)$."

Comment from Reviewer 9: "The presentation of the results continues the deficits of the M&M section. It does not fulfill the existing standards."

Response from Authors 9: We have presented our results in a manner common to model-data fusion studies.

Proposed changes to Manuscript 9: Without further explanation of the reviewer's existing standards we cannot improve this section to their satisfaction.

Comment from Reviewer 10: "What was the quantitative propagation of the initial parameter setting?"

Response from Authors 10: As explained above, the fitting method does not require initial parameter settings and in any case it is not clear to us what propagation of parameter settings refers to. We have striven to offer a clear explanation of the Bayesian fitting method used in our study but given the length limitations of a scientific paper we found that a detailed explanation of the basics of model fitting methods was not feasible.

Proposed changes to Manuscript 10: As in our reply to the other reviewer, we have expanded on our methods paragraph describing how we propagate parameter uncertainty

"To calculate uncertainty for the model predictions we sample parameter values from their respective posterior distribution and compute predictions with each parameter combination, which results in a corresponding distribution of model predictions. We report this prediction distribution uncertainty using 95th percent confidence intervals. This predicted distribution does not include the prescribed or inferred uncertainty about observations, $\sigma$x,D, our predicted distributions correspond to the state being predicted and not the observations of that state."

Comment from Reviewer 11: "This leads to my main criticism of the paper: the results given are not reproducible."

Response from Authors 11: In accordance to the GMD publication requirements, the model code and settings are available upon request from the authors. The model fitting algorithm, developed by our group, has been freely available for several years. All the data used is freely available and fully referenced in the text.

Proposed changes to Manuscript 11: We do not propose any changes because we already include statements about the code and data availability in the manuscript.

---

## Author Comment (AC3) · 2 Feb 2017

We thank both reviewers for their time and commitment in providing us with a great set of constructive advice. We have responded to all of their points of advice in our individual replies to their posted comments. We are confident that doing so has improved our manuscript and will help enable future researchers to build on what we have done.

---

## Author Response (AR2)

Dear Christoph Müller,

Thank you for your handling of this review process. We have been very impressed by your editorial handling. The remaining reviewers comments are very good and we've responded to them fully. We'll detail what we did here and include a marked up version of the manuscript to help you see every change. In summary, we have addressed the remaining issue of referee #3 and expanded on the description of the details of our study as requested by referee #2.

For Reviewer #3, who said:

> " My initial concerns with this paper have all been addressed satisfactorily, with one partial exception. The authors do not include residual variance when estimating prediction uncertainty, explaining that they are interested in predicting true as opposed to observed values. This assumes that residual variance is due solely to measurement error, which is almost certainly not the case for very simplified models of complex systems. The residual error is probably, to a very large extent, due to model error and thus will also contribute to prediction error. This should be acknowledged.

> Other than that, there are some typing errors that should be corrected: P4L6 "description discussion", P11L3 "that that", P13L33 "can" should be "could", P14L16 "begin" should be "being"."

we have included a statement in the methods:

> "Generally, we assume that the variation in the model predictions about the data is solely due to uncertainty in the data. We address the limitations of this assumption and future improvements in the Discussion.",

then in the Discussion we state.

> "Our estimates of model parameter uncertainty, and consequently model prediction uncertainty, are influenced by our assumption that the model is correct and that any departure of the data from predictions is due to measurement error. This is undoubtedly false but makes our parameter estimation method simpler. Overall prediction uncertainty can be decomposed into initial condition uncertainty, parameter uncertainty and model uncertainty and methods exist for making these uncertainty estimates and building them into predictions (Wallach et al., 2016b, a). Such estimates should be made if our model is applied to real agricultural prediction scenarios."

We have added the two additional references cited above and also corrected the minor typos that he noticed.

For Reviewer #2 who said

> " The authors had obviously problems to reply to my critics in respect to the 'reproducibility'. So I will elaborate on this point a little bit more explicit. For reproducing the work, I would expect a detailed description of the equations (no f of something), a table with the original parameter values and the intervals set around them. I assume the authors started with an initial parameter setting and set the intervals around them, which could be done in a general way, but has to be described. If you have freely chosen the intervals, you have to motivate each of them. Furthermore, I would like to see the

expected values of all parameter distributions and how they shifted from the original ones."

and in relation to your own recommendations of

"I would ask you to mainly spill out the details of equation 2, which is currently presented as Gleaf = f(some parameters and other functions)-Rplant. As GMD is dedicated to supply all the page space to fully describe models, I concur with referee #2 that a more explicit description of your model is desirable. I see that the Guilbaud et al. 2014 also did not describe the model in more detail and that the equations have been at least partially described elsewhere, but still the detail of model description is insufficient here. You may want to expand the full set of equations in an appendix or in the paper, as you see fit. Secondly, I would like to see an extra column in table 2 with the pre-defined parameter space used in your study and (if possible) any justification or reference for that."

we have expanded on the previously inadequately explained equations in the manuscript. For brevity we won't paste them here – they are clear in the marked up version. In particular we now include a new appendix (B) that outlines full details of our photosynthesis model. We have also added columns to table 2 as you describe and include a statement in the methods about how these ranges were chosen

"Parameter ranges were set based on literature and our understanding of plausible biological ranges for these crop species and agricultural scenarios as well as additional adjustment to ensure parameter convergence during inference."

We thank both reviewers and yourself for your time and attention to detail.

Sincerely,

Matthew Smith

[revised manuscript text omitted]

---

## Author Response (AR3)

Dear Christoph Müller,

Thank you for picking up those two issues. We have made the corrections you requested. The radiation coefficients are unitless because LAI is itself a simple ratio, however we have indicated that it is unitless in the text.

Thank you sincerely again.

Matthew